**Article** https://doi.org/10.1038/s41467-024-55510-5

# Moesin integrates cortical and lamellar actin networks during *Drosophila* macrophage migration

Besaiz J. Sánchez-Sánchez [1], Stefania Marcotti [1], David Salvador-Garcia [1], María-del-Carmen Díaz-de-la-Loza [1], Mubarik Burki [1], Andrew J. Davidson [2], Will Wood [3] & Brian M. Stramer [1] ✉

Cells are thought to adopt mechanistically distinct migration modes depending on cell-type and environmental factors. These modes are assumed to be driven by mutually exclusive actin cytoskeletal organizations, which are either lamellar (flat, branched network) or cortical (crosslinked to the plasma membrane). Here we exploit *Drosophila* macrophage (hemocyte) developmental dispersal to reveal that these cells maintain both a lamellar actin network at their cell front and a cortical actin network at the rear. Loss of classical actin cortex regulators, such as Moesin, perturb hemocyte morphology and cell migration. Furthermore, cortical and lamellipodial actin networks are interregulated. Upon phosphorylation and binding to the plasma membrane, Moesin is advected to the rear by lamellar actin flow. Simultaneously, the cortical actin network feeds back on the lamella to help regulate actin flow speed and leading-edge dynamics. These data reveal that hemocyte motility requires both lamellipodial and cortical actin architectures in homeostatic equilibrium.

Cell migration is thought to involve characteristically distinct modes of motility consisting of unique actin network architectures. Mesenchymal modes of migration involve firm adherence to the substrate with a flat, branched actin protrusion (i.e., lamella) at the leading edge. In contrast, amoeboid motility requires low substrate attachment and strong cortical acto-myosin contractility around the cell, leading to an increase in hydrostatic pressure that drives bleb-based protrusions at the front[1]. These migration modes are also hypothesized to involve distinct regulatory mechanisms. Ezrin, Radixin, Moesin (ERM) family members, which control cortical actin-membrane crosslinking, are critical for the migration of several cell types thought to undergo amoeboid migration, such as zebrafish germ cells[2], T-cells[3], and neutrophils[4,5]. However, in recent years some of these defining characteristics of supposedly distinct migration modes have started to become blurred. Flat lamellar-like structures are observed in cells

hypothesized to undergo amoeboid motility[6-8], and vice versa, blebs also exist in cells undergoing mesenchymal migration[9,10]. Additionally, ERM proteins, and their localized depletion from the cortex, can drive lamella protrusion and control directionality in cells undergoing mesenchymal migration[10,11]. Data also suggest that regardless of the migration mode, many cell types have both a distinct cortical actin network polarized towards the rear of the cell while also containing a lamellipodia at the front[12,13]. Therefore, opposing gradients of lamellar and cortical actin networks may coexist in migrating cells and it will be critical to understand how they are coordinated to control cell motility.

Embryonic *Drosophila* macrophages (hemocytes) disperse throughout the embryo with broad flat lamellae at their leading edges. Exploiting our capacity to live image lamellar actin flow during hemocyte dispersal, we previously revealed that their movement is

[1]Randall Centre for Cell and Molecular Biophysics, King's College London, SE1 1UL London, UK. [2]Wolfson Wohl Cancer Research Centre, School of Cancer Sciences, University of Glasgow, Garscube Estate, Switchback Road, Bearsden, G61 1BD Glasgow, UK. [3]Centre for Inflammation Research, Institute for Regeneration and Repair, University of Edinburgh, 5 Little France Drive, Edinburgh Bioquarter, EH16 4UU Edinburgh, UK. ✉e-mail: brian.m.stramer@kcl.ac.uk

driven by a highly coordinated retrograde flow of actin from the front to the rear of their lamellar networks[14]. Here, we show that this actin flow transitions into a membrane-associated, cortical actin architecture, revealing that lamellar and cortical actin networks must be coordinated to control hemocyte migration.

## Results

### Hemocytes maintain both lamellipodial and cortical actin networks during their developmental dispersal

Live imaging hemocyte migration reveals a consistent and coordinated retrograde flow of actin from the leading edge of their lamella[14,15]. However, an analysis of the negative divergence in this flow field, which highlights sinks in the actin network (i.e., regions where actin filaments are undergoing contraction and disassembly)[14], along with deformation mapping, revealed a region at the rear of the lamella where the actin network is suddenly becoming compressed (Fig. 1a–c, Supplementary Movie 1). This region of actin compression occurred adjacent to the hemocyte cell body, which contains a dense, surrounding network of actin filaments[14] (Fig. 1b, Supplementary Movie 2). This actin transition also correlated with a change in hemocyte morphology; whereas the lamella is a flat 2-dimensional structure, the cell body is more spherical (Fig. 1d, Supplementary Movie 3). Interestingly, this region of increased actin compression correlated with the accumulation of classical actin cortex-associated components such as Moesin and phosphatidylinositol 4,5-bisphosphate (PIP$_2$)[16], suggesting the presence of a distinct cortical actin network, which may be driving the spherical alteration in cellular morphology due to an increase in cortical tension around the plasma membrane (Fig. 1e, Supplementary Movie 4). We therefore adapted a membrane proximal F-actin probe for flies (MPAct), which was previously used to highlight distinct cortical actin networks in mammalian cells in vitro[12]. In brief, we generated fluorescently-tagged transgenes expressing an actin binding domain containing a membrane-targeting sequence (MPAct), along with control probes consisting of the membrane or actin binding proteins alone. Therefore, when MPAct and control probes are co-expressed in cells, ratiometric analysis can reveal regions of the cell containing distinct actin filaments associated with the plasma membrane[12]. Expression of this probe in follicular, gut, and salivary gland epithelial cells, as well as border cells, all of which are reported to have a Moesin-regulated apical actin cortex[17–21], revealed MPAct enrichment at the apical cortex, highlighting that the MPAct probe can indeed identify distinct actin networks in cells (Supplementary Fig. 1). Expression of this probe in hemocytes also revealed a distinct actin network in close proximity to the plasma membrane at the rear of lamellae surrounding their cell bodies, confirming the presence of an actin cortex. These results reveal that lamellar and cortical actin networks coexist in migrating hemocytes (Fig. 1f, Supplementary Movie 5, 6).

### Moesin is required for proper hemocyte morphology and developmental dispersal

We next examined whether hemocyte morphology was perturbed in the absence of Moesin. Embryos containing a mutant allele of *Drosophila moesin*, the only ERM family member in flies, showed a decrease in actin cortical enrichment, accompanied with an increase in cell body volume, and a decrease in cell body sphericity, while lamellar size was unaffected (Fig. 2a, b). Furthermore, there was an increase in negative divergence values in the actin flow field surrounding the cell body suggesting a reduction in the rate of compression of the actin network (Fig. 2c, d). Finally, analysis of the MPAct probe revealed that *moesin* mutant hemocytes showed a reduction in a membrane-associated actin network around their cell bodies, which is consistent with the reduction in cortical actin (Fig. 2e, f, Supplementary Movie 7).

We subsequently examined the migratory capacity of hemocytes upon perturbation of Moesin. *Moesin* mutant embryos showed defects in hemocyte developmental dispersal and a reduction in cell speed and

persistence (Fig. 3a–c, Supplementary Movie 8). Additionally, hemocyte repulsion upon collision with neighbors (contact inhibition of locomotion, CIL) was also perturbed in the absence of Moesin (Fig. 3d), suggesting that regulation of the cortical actin network is required for normal CIL dynamics. Embryonic dispersal defects were also phenocopied by macrophage-specific expression of an RNAi for Moesin, suggesting that these *moesin* mutant migration defects were cell-autonomous (Fig. 3e-g). Moesin association with the actin cortex is regulated by phosphorylation, and embryos mutant for sterile20-like kinase (*slik*), the kinase responsible for Moesin activation during mitotic rounding[22,23], also had macrophage dispersal defects. Additionally, macrophage-specific expression of an RNAi for slik showed a similar phenotype to *moesin* mutants (Fig. 3e, g). Furthermore, Moesin activity during mitosis has been shown to be modulated by PIP$_2$ binding[24]; embryos mutant for the Phosphatidylinositol 4-phosphate 5-kinase, *skittles* (*sktl*), and the Phosphatidylinositol 3-phosphatase (*Pten*), which are both involved in PIP$_2$ production[24], showed hemocyte migration defects that phenocopy loss of Moesin (Fig. 3e–g). These data reveal that, similar to Moesin's involvement in controlling cell rounding during mitosis[23], Moesin is also required to control actin cortex mechanics in migrating hemocytes, which is needed for their normal cell body morphology and motility.

### Moesin phosphorylation and PIP$_2$ binding must be regulated for normal hemocyte migration

Moesin activity is controlled by the cooperative action of phosphorylation of a threonine residue in the C-terminal actin-binding domain and association with the plasma membrane lipid, PIP$_2$[25] (Fig. 4a). We, therefore, investigated the activity of a series of *Drosophila moesin* transgenes containing either the full length wild-type protein or mutations in the phosphorylation and PIP$_2$ binding sites[26] (Fig. 4b). Expression of single copies of these GFP-tagged transgenes in a wild-type background specifically in hemocytes did not cause severe defects in dispersal (Supplementary Fig. 2). However, expression of two copies of several transgenes (i.e., two copies of both Gal4 and UAS transgenes) led to a severe effect on hemocyte embryonic dispersal. Two copies of a Lysine to Asparagine (K/N) mutation (Moesin$^{KN}$), which affects Moesin's PIP$_2$ binding capacity, induced defects in hemocyte dispersal in a small proportion of embryos (Figs. 3e, 4c). In contrast, two copies of transgenes containing the K/N mutation together with a T/A mutation, which prevents Moesin phosphorylation (Moesin$^{TAKN}$), led to more severe dispersal defects, suggesting functional cooperativity between PIP$_2$ association and Moesin phosphorylation (Figs. 3e, 4c). Additionally, two copies of a phosphomimetic Moesin mutant (T/D, Moesin$^{TD}$), which is hypothesized to be constitutively active, led to a majority of embryos containing developmental dispersal defects, revealing that over-activation of Moesin is severely detrimental for motility (Figs. 3e, 4c). Interestingly, addition of the PIP$_2$ binding mutation to the constitutively active, phosphomimetic mutant (Moesin$^{TDKN}$) rescued the hemocyte dispersal defects (Figs. 3e, 4c). These data therefore suggest that both Moesin phosphorylation and PIP$_2$ binding are required for full Moesin activity in migrating hemocytes.

We subsequently investigated the effects of Moesin hyperactivation on hemocyte morphology and the cortical actin network (Fig. 4d, e). While cell body sphericity was unaffected, Moesin$^{TD}$ showed an increase in cell body volume and a reduction in lamellar volume compared to wild-type Moesin (Moesin$^{WT}$), suggesting that the increase in the size of the hemocyte soma came at the expense of the lamellar network (Fig. 4d, e). Furthermore, Moesin$^{TD}$ led to an overall increase in cell sphericity, which is likely explained by the loss of the planar lamella (Fig. 4d, e). Additionally, analysis of the actin cortex with MPAct revealed that while there was not an overall change in the amount of cortical actin, Moesin$^{TD}$ led to an alteration in cortical distribution (Fig. 4f–h, Supplementary Movie 9). In wild-type cells, there was a graded distribution of cortical actin from the front to the rear of

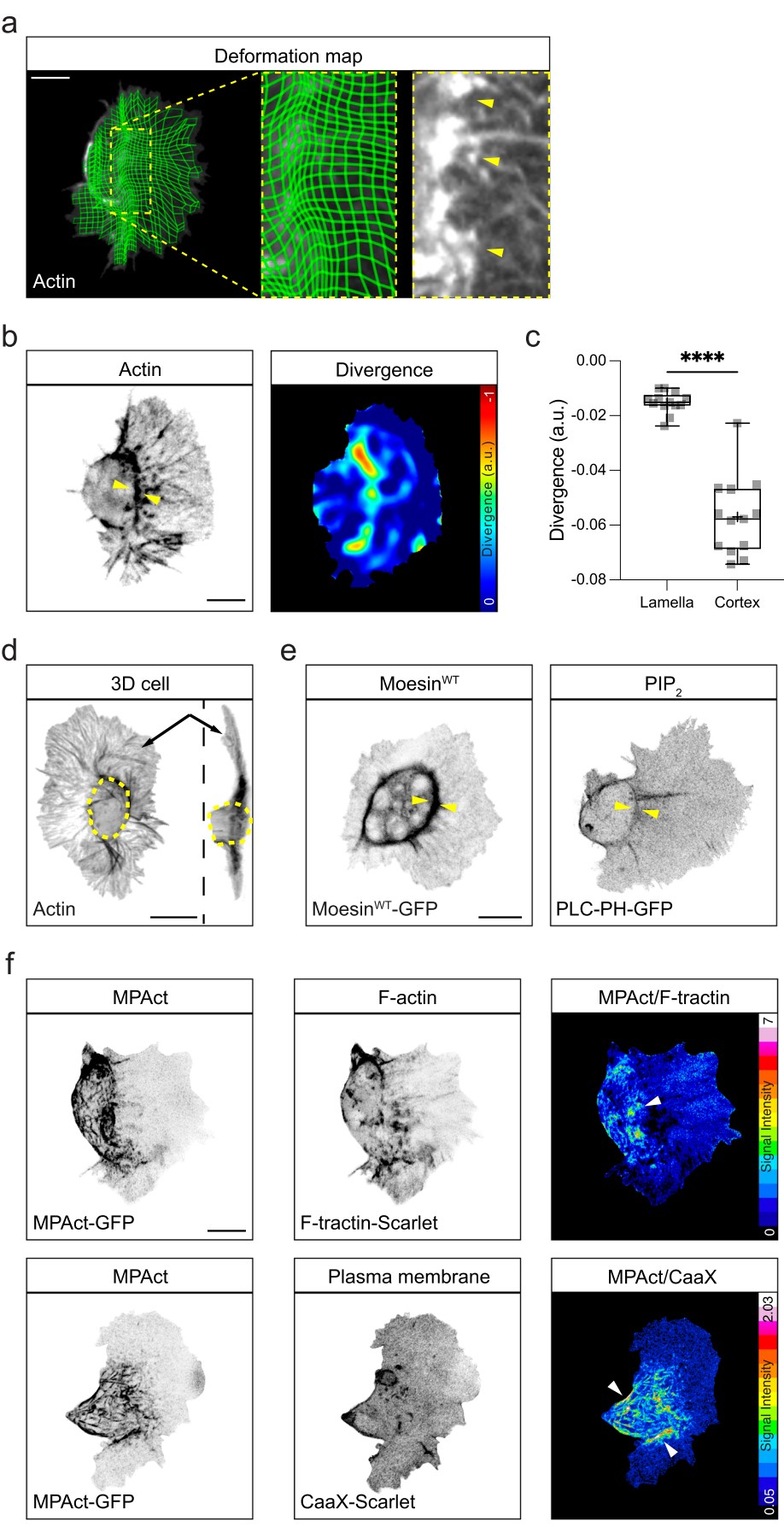

**Fig. 1 | Hemocytes contain both a lamellar and cortical actin network.**
**a** Deformation map of the lamellar actin retrograde flow field highlighting the increase in actin network deformation around the cell body (arrowheads).
**b** Analysis of the negative divergence in the actin retrograde flow field revealing network compression and actin accumulation around the hemocyte cell body (arrowheads). Only negatively divergent regions are highlighted. a.u., arbitrary units. **c** Quantification of negative divergence at the lamellar and cortical regions. ****$P$ < 0.0001, Mann–Whitney two-tailed test. Boxplot shows medians, 25th and 75th percentiles as box limits, minimum and maximum values as whiskers; each

datapoint is displayed as a marker ($n$ = 13 hemocytes for each genotype). **d** 3D reconstruction of a hemocyte highlighting the transition between the flat lamellar network (arrows) and the rounded cell body (dashed circle). **e** Localization of Moesin and PIP$_2$ (PLC-PH-GFP) to the region surrounding the hemocyte cell body (arrowheads). **f** Ratiometric imaging of a membrane-proximal actin probe (MPAct) with an actin (F-tractin) or membrane probe (CaaX) highlighting an increase in membrane-associated actin around the hemocyte cell body (arrowheads). Scale bars, 10 μm.

the lamella with a peak around the cell body (Fig. 4f, h, Supplementary Movie 9). However, Moesin$^{TD}$ led to a flattening of the MPAct gradient suggesting enhanced association of the actin network with the plasma membrane within the lamella (Fig. 4f, h, Supplementary Movie 9). These data suggest that premature formation of cortical actin within the lamellar region inhibits lamellar extension and alters the balance of lamellar and cortical actin networks.

## Moesin localization and formation of the actin cortex are regulated by the lamellar actin flow

We subsequently examined the localization of GFP-tagged Moesin mutant transgenes to understand how Moesin regulation may be controlling its cortical distribution. In this case, we expressed single copies of these transgenes in a wild-type background as this does not lead to significant defects in hemocyte motility (Supplementary Fig. 2). While most of the transgenes showed some enrichment around the cell body, the phosphomimetic form, Moesin$^{TD}$, led to a significant increase in cortical localization (Fig. 5a, b, Supplementary Movie 10). Consistent with our previous data revealing cooperativity of phosphorylation and PIP$_2$ binding, addition of the K/N mutation to the phosphomimetic form of Moesin (Moesin$^{TDKN}$) reduced the cortex-specific localization (Fig. 5a, b). We also observed subtle differences in Moesin localization in the lamella; Moesin$^{TD}$ showed a gradient of localization that extended from the cell body into the lamella, suggesting that it is also progressively interacting with actin filaments towards the rear of the lamellar network (Fig. 5a, c, Supplementary Movie 10). Consistent with these data, live imaging of actin and Moesin constructs revealed that phosphomimetic Moesin flows in a retrograde fashion, which is most highly correlated with the direction of actin flow compared to the other *moesin* mutant transgenes (Fig. 6a, b, Supplementary Movie 10). Addition of the K/N mutation to the phosphomimetic form (Moesin$^{TDKN}$) reduced the graded distribution of Moesin in lamellae and reduced its correlated motion with the actin retrograde flow (Figs. 5c, 6b). These data suggest that cooperative interactions between Moesin phosphorylation and PIP$_2$ binding lead to Moesin advection by the flowing actin network to the rear of lamellae in hemocytes.

Previous work in migrating cells revealed that any protein with a sufficient actin-binding capacity is prone to advection by the actin retrograde flow leading to a graded distribution towards the rear of the lamellar network[27,28]. We, therefore, analyzed the dynamics of Moesin transgenes by fluorescence recovery after photobleaching (FRAP) to examine alterations in diffusion, which could suggest changes in actin-binding efficiency. FRAP analysis revealed that Moesin$^{TD}$ showed an increased half life in lamellae compared to other mutant transgenes (Fig. 6c, d). The addition of the K/N mutation to Moesin$^{TD}$ (i.e., Moesin$^{TDKN}$) increased its diffusivity, suggesting that PIP$_2$ binding is also required to fully reduce the diffusion of activated Moesin (Fig. 6c, d). These data indicate that a combination of phosphorylation and PIP$_2$ binding leads to a reduction in Moesin mobility in the lamellar network through enhanced actin binding and/or increased association with the plasma membrane.

Recent work has revealed that increased association with the plasma membrane can enhance advection of molecules by actin flow[29]. We therefore examined the effects of artificially altering the

membrane-associating capacity of Moesin$^{TDKN}$, which has reduced hemocyte migratory defects compared to Moesin$^{TD}$ (Fig. 4c). Morphotrap internal and Morphotrap external are a set of *Drosophila* transgenes that recruit GFP-tagged proteins to the internal or external face of the plasma membrane, respectively[30]. Recruiting Moesin$^{TDKN}$ to the external plasma membrane (Morphotrap external) showed no defects in hemocyte developmental dispersal, similarly to Moesin$^{TDKN}$ alone (Figs. 4c, 6e, f). However, recruitment to the internal plasma membrane (Morphotrap internal) led to severe migration defects, mimicking the phenotype of Moesin$^{TD}$, which suggested that association with the membrane is required for activated Moesin to function during hemocyte motility (Figs. 4c, 6e, f). These data suggest that efficient Moesin advection by the actin retrograde flow requires actin-binding and membrane association, which drive Moesin to the rear of lamellae to regulate the cortical actin network surrounding the cell body.

## Moesin activity is required for normal lamellar actin flow and leading edge dynamics

While these data reveal that Moesin regulates the actin cortex at the rear of migrating hemocytes, we also wondered whether Moesin may be functionally required for lamellar network activities. An analysis of hemocyte morphodynamics revealed that in the absence of Moesin, hemocytes showed a reduction in lamellar polarity (Fig. 7a, b, Supplementary Movie 11). Additionally, loss of Moesin led to a decrease in leading edge velocity suggesting feedback with the protrusion machinery at the cell front (Fig. 7c, d). Furthermore, there was a reduction in leading edge protrusions that correlate with hemocyte motion suggesting a reduction in migratory efficiency[14,31], which may explain the alteration in cell persistence (Figs. 3c, 7e, f). *Moesin* mutants, and two copies of the phosphomimetic *moesin* transgene, also showed a significant reduction in actin retrograde flow speed, suggesting that Moesin activity must be precisely controlled for normal lamellar actin flows (Fig. 7g–j). As Myosin is also required for actin retrograde flow and cortex organization[14,32,33], we determined whether Moesin may be regulating Myosin activity or localization. However, we observed no change in the cortical enrichment of Myosin in *moesin* mutant hemocytes (Supplementary Fig. 3a, b). Additionally, live imaging revealed that while active Moesin (Moesin$^{TD}$) decorated most of the cortex, Myosin was more punctate (Supplementary Fig. 3c). Finally, examination of actin flow organization by streamline analysis revealed that myosin mutant cells had a much more disrupted actin flow field than *moesin* mutant hemocytes (Supplementary Fig. 3d), suggesting that Myosin and Moesin have distinct functions in hemocytes. These data reveal that the hemocyte actin cortex at the rear of the cell or a progressive increase in Moesin activity within the lamella is needed to control migratory behaviors at the cell front.

## Discussion

Here we reveal that *Drosophila* hemocyte migration requires the coexistence of both lamellar and cortical actin networks. Upon reduction of the hemocyte cortex, the cell body expands and flattens suggesting that cortical tension is altered. As hemocyte migration is severely confined during their developmental dispersal between the epithelium and ventral nerve cord[34], a rigid cortex may be essential to

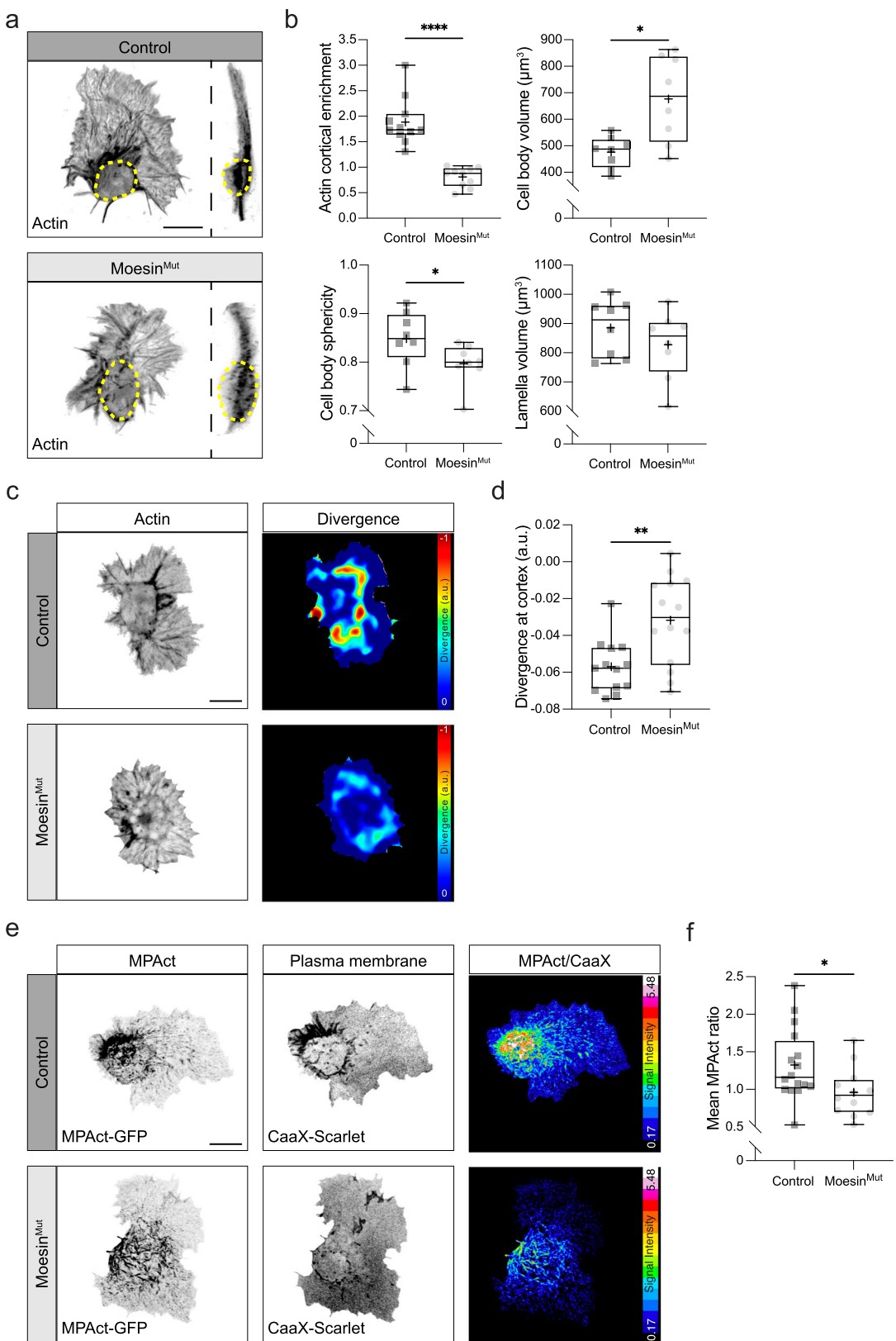

prevent the cell from becoming overly deformed, which may partly explain the developmental dispersal failure in the absence of Moesin. However, the pleiotropic effects on lamellar morphology and dynamics as a result of alterations in cortex regulation suggest that there is complex interregulation occurring between these two actin networks, which is needed to control front-back coordination during cell motility.

ERM family members are stereotypical cortical regulators that are localized to the rear of many migrating cell types[2–5]. Acto-myosin flow has emerged as a critical player in coordinating migratory behaviors across the cell due to its capacity to advect regulatory factors[2,14,27,32], and our data suggest that ERM protein localization to the rear of migrating cells can be explained by this transport mechanism. Ezrin, an ERM family protein, was recently observed to be transported by actin

**Fig. 2 | Loss of Moesin perturbs the cortical actin network and alters the morphology of the hemocyte cell body. a** 3D reconstruction of a Control and Moesin^Mut hemocyte highlighting the increase in cell body size in the absence of Moesin (dashed circle). **b** Quantification of actin enrichment around the cortex, cell body volume, sphericity, and lamellar volume. ****$P$ < 0.0001 (actin cortical enrichment), *$P$ = 0.01 (cell body volume), 0.04 (sphericity), ^ns$P$ = 0.5 (lamellar volume). Mann–Whitney two-tailed tests. (actin cortical enrichment: $n$ = 11 Control and 10 Moesin^Mut hemocytes; cell body volume, sphericity and lamellar volume: $n$ = 8 hemocytes for each genotype). **c** Analysis of the negative divergence in the actin retrograde flow field in Control and Moesin^Mut hemocytes (a.u., arbitrary

units). **d** Quantification of negative divergence at the cortex surrounding the hemocyte cell body revealing a reduction in actin network compression. **$P$ = 0.0056, Mann–Whitney two-tailed test. ($n$ = 13 Control and 14 Moesin^Mut hemocytes). **e** Ratiometric analysis of MPAct in Control and Moesin^Mut hemocytes. **f** Quantification of MPAct levels in Control and Moesin^Mut hemocytes. *$P$ = 0.015, Mann–Whitney two-tailed test. ($n$ = 16 Control and 12 Moesin^Mut hemocytes). Scale bars, 10 μm. All boxplots show medians, 25th and 75th percentiles as box limits, minimum and maximum values as whiskers; each datapoint is displayed as a marker. Scale bars, 10 μm.

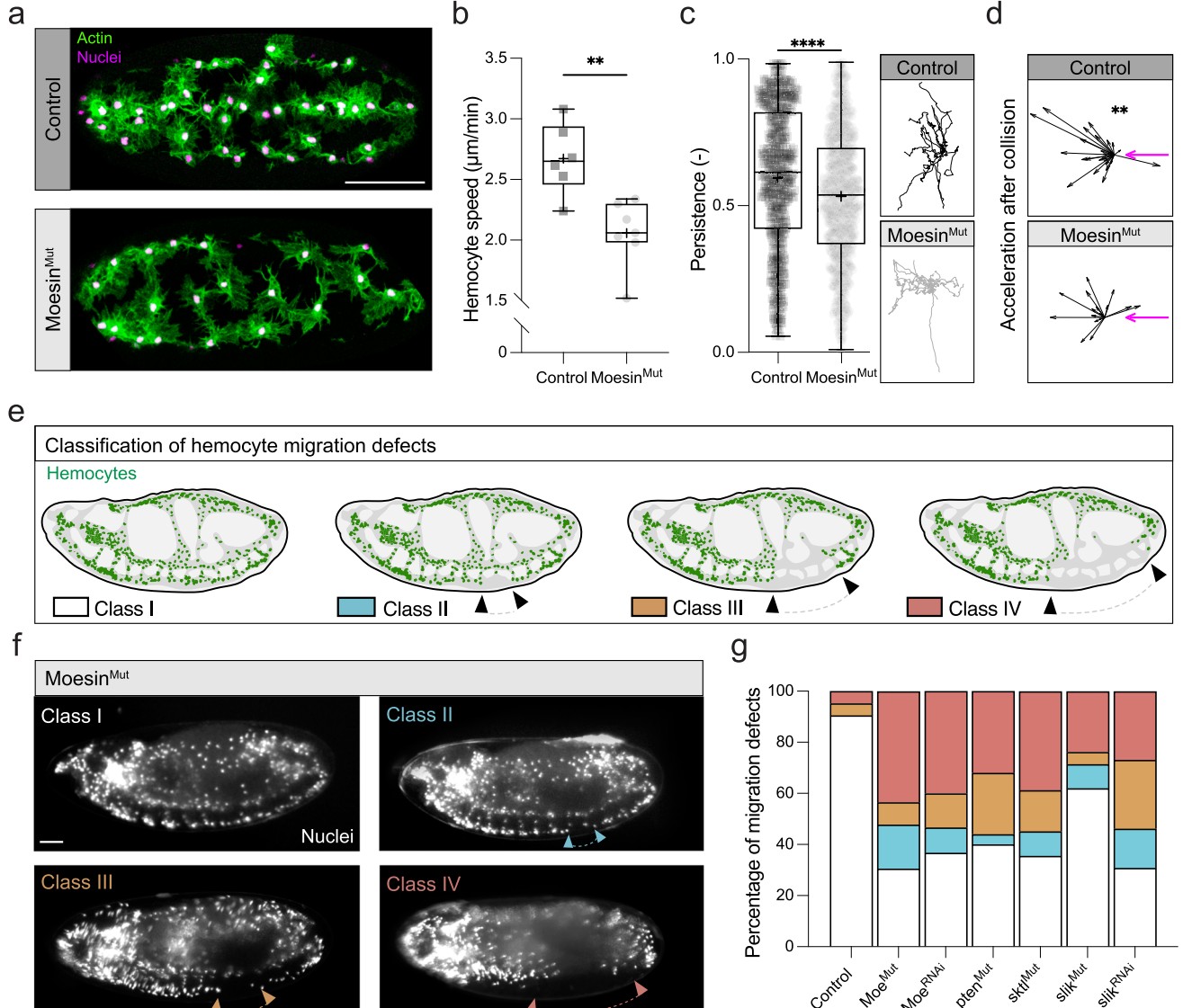

**Fig. 3 | Moesin and its regulators are required for hemocyte developmental dispersal. a** Imaging of Control and Moesin^Mut hemocytes on the ventral surface of the embryo revealing perturbation of hemocyte dispersal in the absence of Moesin. **b** Quantification of hemocyte speed in Control and Moesin^Mut hemocytes. **$P$ = 0.0047, Mann–Whitney two-tailed test. ($n$ = 457 Control and 229 Moesin^Mut hemocytes from 6 and 7 embryos, respectively). **c** Nuclear tracks displayed normalized to a common starting point and quantification of hemocyte persistence in Control and Moesin^Mut hemocytes. ****$P$ < 0.0001, Mann–Whitney two-tailed test. ($n$ = 537 Control and 844 Moesin^Mut samples from 12 and 14 hemocytes, respectively) **d** Quantification of the change in hemocyte acceleration upon contact inhibition of locomotion. Magenta arrow highlights the position of the colliding partner. **$P$ = 0.0015, ^ns$P$ = 0.3203 Wilcoxon signed-rank test (hypothetical median

value of 0, $n$ = 23 Control and 11 Moesin^Mut collision events). **e** Schematic highlighting the classification of the degree of hemocyte developmental dispersal defects based on how far hemocytes have migrated along the ventral nerve cord (arrowheads). **f** Analysis of hemocyte migration in Moesin^Mut embryos revealing the four classes of dispersal defects. **g** Quantification of the degree of hemocyte dispersal defects in Moesin^Mut embryos, embryos expressing Moesin RNAi specifically in hemocytes, and embryos mutant for Moesin regulators, pten, sktl, slik, and embryos expressing slik RNAi specifically in hemocytes ($n$ = 21 Control, 23 Moesin^Mut, 30 Moesin RNAi, 25 pten^Mut, 31 sktl^Mut, 21 slik^Mut and 26 slik RNAi embryos). Scale bars, 50 μm. All boxplots show medians, 25th and 75th percentiles as box limits, minimum and maximum values as whiskers; each datapoint is displayed as a marker. Scale bars, 10 μm.

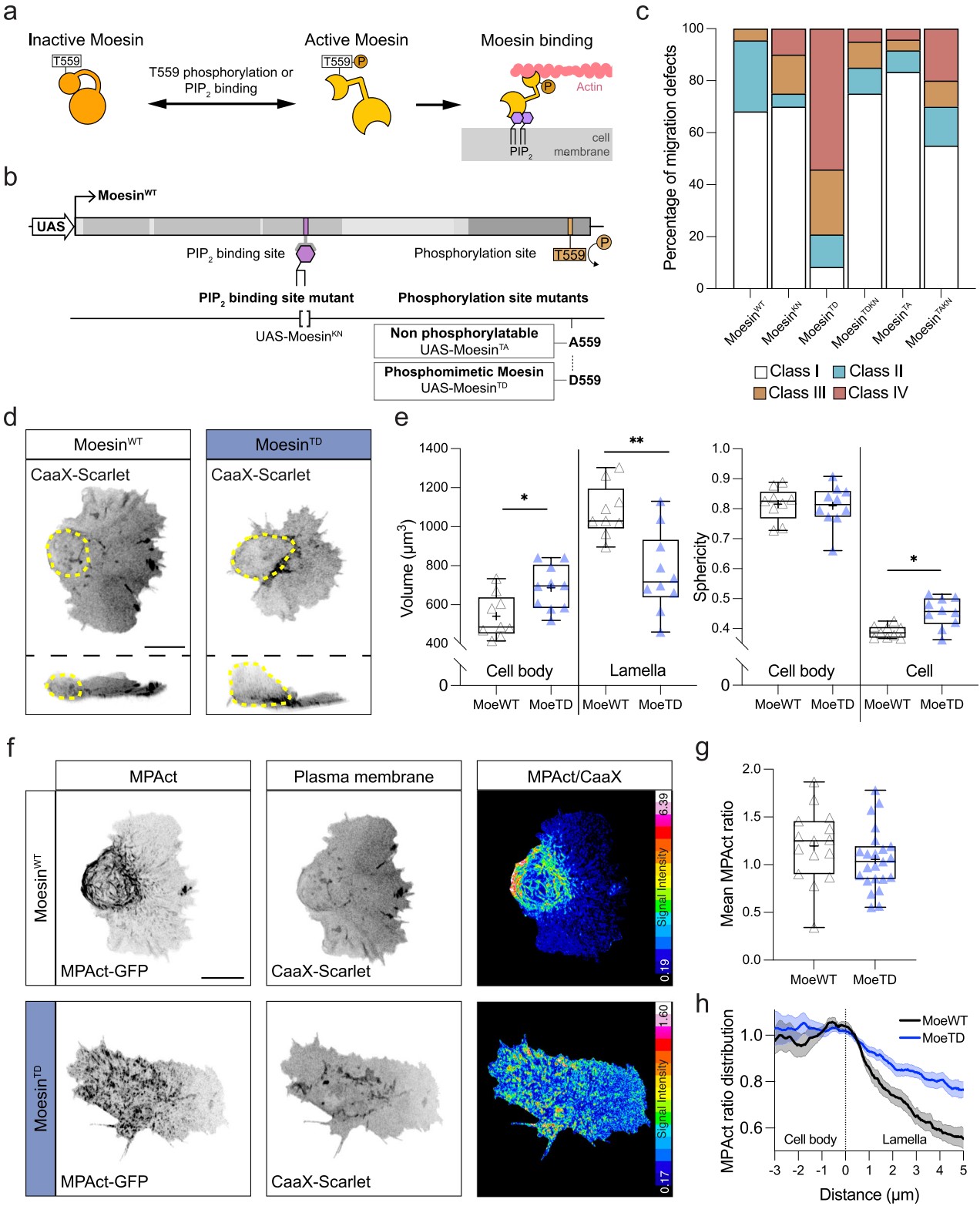

flow in migrating zebrafish germ cells, however, germ cell migration is thought to be mostly, if not entirely, amoeboid in nature driven, by bleb-based protrusions at the front and cortical actin flows around the plasma membrane. Moesin advection in hemocytes requires interactions with a lamellar actin network suggesting that Moesin may have migratory roles beyond amoeboid motility. Additionally, recent observations highlighting that cells simultaneously maintain cortical and lamellar actin networks[12,13], and that cortical regulators, such as Moesin, have roles even in cells with lamellae[11] (this work) suggest that

dividing migration into binary modes (i.e., amoeboid vs mesenchymal) may be overly simplistic.

Activation of ERM proteins through phosphorylation and PIP$_2$ binding has been hypothesized to occur in a two-step process: first, PIP$_2$ binding induces a partial conformational change, which subsequently makes the phosphorylation site accessible to kinase activation allowing for actin-binding[25]. However, contrary to previous work in epithelial cells suggesting that phosphorylation of Ezrin renders it independent of PIP$_2$[35], our data suggest that in

**Fig. 4 | Moesin phosphorylation and PIP$_2$ binding are required for Moesin function in regulating hemocyte migration and cell morphology. a** Schematic highlighting the molecular mechanisms involved in the regulation of Moesin activity. **b** Schematic of the domain organization of Moesin and the *Drosophila* transgenic constructs containing point mutations in key residues that control Moesin activity. **c** Quantification of the degree of developmental dispersal defects when two copies of the transgenic Moesin constructs are driven specifically in hemocytes ($n$ = 22 Moesin$^{WT}$, 20 Moesin$^{KN}$, 24 Moesin$^{TD}$, 20 Moesin$^{TDKN}$, 24 Moesin$^{TA}$ and 20 Moesin$^{TAKN}$ embryos). **d** 3D reconstruction of a hemocyte expressing Moesin$^{WT}$ or a Moesin$^{TD}$ transgenes revealing an increase in the size of the hemocyte cell body (dashed circle) when Moesin is constitutively active. **e** Quantification of cell body and lamellar volume, and cell body and overall cell sphericity in hemocytes expressing Moesin$^{WT}$ or Moesin$^{TD}$. *$P$ = 0.0168 (cell body volume),

**$P$ = 0.0076 (lamellar volume), $^{ns}P$ = 0.9682 (cell body sphericity), *$P$ = 0.0101 (cell sphericity), Mann–Whitney two-tailed test. ($n$ = 9 Moesin$^{WT}$ and 10 Moesin$^{TD}$ hemocytes). **f** Ratiometric analysis of MPAct in hemocytes expressing Moesin$^{WT}$ or Moesin$^{TD}$. **g** Quantification of MPAct levels in hemocytes expressing Moesin$^{WT}$ or Moesin$^{TD}$ revealing no overall alteration in membrane-associated actin. $^{ns}P$ = 0.1214, Mann–Whitney two-tailed test. ($n$ = 15 Moesin$^{WT}$ and 23 Moesin$^{TD}$ hemocytes). **h** Line scan analysis of MPAct distribution in hemocytes expressing Moesin$^{WT}$ or Moesin$^{TD}$ normalized to the cell body/lamellar transition highlights an increase in membrane-associated actin within the lamellar region when Moesin is constitutively active; data are presented as mean values +/- standard error of the mean ($n$ = 15 Moesin$^{WT}$ and 23 Moesin$^{TD}$ hemocytes). Scale bars, 10 μm. All boxplots show medians, 25th and 75th percentiles as box limits, minimum and maximum values as whiskers; each datapoint is displayed as a marker. Scale bars, 10 μm.

hemocytes, continued PIP$_2$ interaction, even after Moesin is phosphorylated, is required for Moesin function during migration. These data are consistent with work in T-cells revealing that phospho-Moesin requires continued PIP$_2$ binding for membrane association[36] and it will be interesting to determine whether these differences are due to cell-type, ERM protein-specific regulatory mechanisms, or whether cell migration has a unique regulatory requirement for continued association with PIP$_2$. Another question is related to the precise role of this lipid interaction. One possibility is that PIP$_2$ binding synergizes with phosphorylation to open the molecule from its autoinhibited state to associate with actin. However, another possibility is that Moesin's interaction with the plasma membrane may decrease its diffusion leading to more efficient advection by the actin flow. It was recently revealed that during polarization of the *C. elegans* zygote, binding to the plasma membrane increases the timescale over which actin flow drives the movement of PAR proteins[29]. Synergy between membrane-binding and actin flow is consistent with our Morphotrap rescue of the Moesin$^{TDKN}$ mutation, suggesting that this mechanism of advection-control is potentially more widespread than initially thought.

The broader effects of Moesin dysregulation on lamellar dynamics suggest that Moesin crosslinking of actin filaments within lamellae during its advection is helping to generate forces that drive actin flow. Additionally, Moesin regulation of the cortical actin network at the rear of the cell may be indirectly aiding the contractile activity within the lamella, which our data suggest is independent of Myosin activity. Related to the latter possibility, there are numerous studies revealing that there are instructive migratory cues coming from the rear of the cell that are needed to control cell polarity and coordinate migratory behaviors, such as membrane tension and cell contractility[37–41], which may be regulated by ERM proteins[42]. These data are also consistent with work in neutrophils revealing that Moesin can aid the directional memory of migrating cells[5]. We hypothesize that this may be controlled by Moesin's capacity to directly or indirectly integrate distinct lamellar and cortical actin architectures at both the front and rear of migrating cells.

## Resources

| REAGENT or RESOURCE | SOURCE | IDENTIFIER |
|---|---|---|
| Experimental Models: Organisms/Strains | | |
| *D. melanogaster:* Sn-Gal4 | 43 | N/A |
| *D. melanogaster:* Srp-Gal4 | 44,45 | N/A |
| *D. melanogaster:* UAS-LifeActGFP | 43 | N/A |
| *D. melanogaster:* UAS-RedStinger | Bloomington *Drosophila* Stock Center | 8547 |
| *D. melanogaster:* srpHemo-Moe::3xmCherry | 46 | N/A |
| *D. melanogaster:* UAS-GMA-GFP | 47 | N/A |

| | | |
|---|---|---|
| *D. melanogaster:* UAS-MPAct-mNeonGreen | This paper | N/A |
| *D. melanogaster:* UAS-CaaX-mScarlet | This paper | N/A |
| *D. melanogaster:* UAS-F-tractin-mScarlet | This paper | N/A |
| *D. melanogaster:* UAS-Sqh-mCherry (Myosin-II light chain) | This paper | N/A |
| *D. melanogaster:* UAS-Zip-GFP (Myosin-II heavy chain) | 48 | N/A |
| *D. melanogaster:* Zip$^1$ (*myosin-II heavy chain mutant*) | Bloomington *Drosophila* Stock Center | 4199 |
| *D. melanogaster:* UAS-PLC-PH-GFP | 49 | N/A |
| *D. melanogaster:* UAS-MoesinRNAi | Vienna *Drosophila* Resource Center | 37917 |
| *D. melanogaster:* UAS-Morphotrap Internal | 30 | N/A |
| *D. melanogaster:* UAS-Morphotrap External | 30 | N/A |
| *D. melanogaster:* UAS-CLIP-GFP | 50 | N/A |
| *D. melanogaster:* UAS-Moesin$^{WT}$-GFP | 51 | N/A |
| *D. melanogaster:* UAS-Moesin$^{TD}$-GFP | 51 | N/A |
| *D. melanogaster:* UAS-Moesin$^{TA}$-GFP | 51 | N/A |
| *D. melanogaster:* UAS-Moesin$^{KN}$-GFP | 26 | N/A |
| *D. melanogaster:* UAS-Moesin$^{TDKN}$-GFP | 26 | N/A |
| *D. melanogaster:* UAS-Moesin$^{TAKN}$-GFP | 26 | N/A |
| *D. melanogaster:* UAS-Moesin$^{WT}$ | 26 | N/A |
| *D. melanogaster:* UAS-Moesin$^{KN}$ | 26 | N/A |
| *D. melanogaster:* UAS-Moesin$^{TD}$ | 26 | N/A |
| *D. melanogaster:* UAS-Moesin$^{TA}$ | 26 | N/A |
| *D. melanogaster:* UAS-Moesin$^{TDKN}$ | 26 | N/A |
| *D. melanogaster:* UAS-Moesin$^{TAKN}$ | 26 | N/A |
| *D. melanogaster:* Moesin$^{PL106}$ | 51 | N/A |
| *D. melanogaster:* Pten$^3$ | 52 | N/A |
| *D. melanogaster:* Slik$^{KG04837}$ | Bloomington *Drosophila* Stock Center | 14435 |
| *D. melanogaster:* UAS-slik RNAi | Bloomington *Drosophila* Stock Center | 35179 |
| *D. melanogaster:* sktl$^{Δ20}$ | Bloomington *Drosophila* Stock Center | 39674 |
| D. melanogaster: Tub-Gal-4 | Bloomington *Drosophila* Stock Center | 5138 |
| Oligonucleotides | | |
| Primer: 5'GGAATTGG-GAATTCGTTTTAATTAAATGGT-GAGCAAGGGCGAG3' | This paper | N/A |
| Primer: 5'GATCCTCTA-GAGGTACCTAGGCTA-CATCAGGCAGCACTTCCT3' | This paper | N/A |
| Primer: 5'GGAATTGG-GAATTCGTTTTAATTAAAATGGG-CATGGCCCGC3' | This paper | N/A |

| | | |
|---|---|---|
| Primer: 5'GATCCTCTA-GAGGTACCTAGGCTACTTGTA-CAGCTCGTCCATGC3' | This paper | N/A |
| **Recombinant DNA** | | |
| Plasmid: pUASt-attB | General Fly Transformation Vectors | 1419 |
| Plasmid: pUASt-attB-CaaX-mScarlet | This paper | N/A |
| Plasmid: pUASt-attB-F-tractin-mScarlet | This paper | N/A |
| Plasmid: pUASt-attB-MPAct-mNeonGreen | This paper | N/A |
| **Software and Algorithms** | | |
| LAS AF | Leica | http://leica-microsystems.com/home/ |
| Zen | Carl Zeiss | https://www.zeiss.com/microscopy/int/products/microscope-software/zen-lite.html |
| Zen Black | Carl Zeiss | https://www.zeiss.com/microscopy/int/products/microscope-software/zen.html#downloads |
| ImageJ/Fiji | Fiji | http://fiji.sc/ |
| Imaris | Bitplane | https://imaris.oxinst.com |
| MATLAB | MathWorks | https://uk.mathworks.com/products/matlab.html |
| Illustrator | Adobe | https://www.adobe.com/uk/products/illustrator.html |
| Prism | GraphPad | https://www.graphpad.com |
| **Other** | | |
| *Drosophila* injection service | BestGene | https://thebestgene.com/ |
| 10S Voltalef oil | VWR | 24627.188 |
| Lumox culture dish | Sarstedt (94.6077.305) | https://www.sarstedt.com/en/products/laboratory/cell-tissue-culture/lumox-technology/product/94.6077.305/ |
| M205 fluorescent dissection microscope | Leica | http://leica-microsystems.com/home/ |
| PLANAPO 2.0x objective for M205 | Leica | 10450030 |
| LSM 880 confocal microscope | Carl Zeiss | https://www.zeiss.co.uk/microscopy/dynamic-content/news/2014/news-lsm-880.html |
| PerkinElmer Ultraview spinning disk Microscope | PerkinElmer | |
| 63x NA 1.4 Plan-Apochromat oil objective for LSM 880 | Carl Zeiss | https://zeiss.com/corporate/int/home.html |
| 40x NA 1.3 Plan-Apochromat oil objective for LSM 880 | Carl Zeiss | https://zeiss.com/corporate/int/home.html |
| 63x NA 1.4 Plan-Apochromat oil objective for Ultraview spinning disk (PerkinElmer) | PerkinElmer | |

## Fly stocks and preparations

Sn-Gal4[43] and Srp-Gal4[44,45] were used to express transgenes specifically in hemocytes. The following UAS lines were used: UAS-LifeActGFP[43], UAS-RedStinger (BDSC, 8547), UAS-GMA-GFP[47], UAS-MPAct-mNeonGreen, UAS-CaaX-mSCarlet, UAS-F-tractin-mScarlet and UAS-Sqh-mCherry (Myosin-II light chain) generated in this paper, UAS-Zip-GFP (Myosin-II heavy chain)[48], UAS-PLC-PH-GFP[49], UAS-Moesin[RNAi] (VDRC, 37917), UAS-Morphotrap Internal and UAS-Morphotrap External (BDSC, 68171 and 68172), UAS-CLIP-GFP[50], UAS-Slik[RNAi] (BDSC, 35179), UAS-Moesin[WT]-GFP, UAS-Moesin[TD]-GFP and, UAS-Moesin[TA]-GFP[51], UAS-Moesin[KN]-GFP, UAS-Moesin[TDKN]-GFP, UAS-Moesin[TAKN]-GFP, UAS-Moesin[WT], UAS-Moesin[KN], UAS-Moesin[TD], UAS-Moesin[TA], UAS-Moesin[TDKN], and UAS-Moesin[TAKN][26]. SrpHemo-Moe::3xmCherry[46] was used to label actin in hemocytes independently of Gal4. The following mutants were used: Moesin[PL106 51], Pten[3,52], Slik[KG04837] (BDSC, 14435), sktl[Δ20] (BDSC, 39674) and Zip[1] *(myosin-II* heavy chain mutant) (BDSC, 4199). TubGal-4 (BDSC, 5138) was used to express UAS-MPAct-mNeonGreen, UAS-CaaX-mScarlet and UAS-F-tractin-mScarlet in other tissues. Flies were transferred to egg-laying cages on grape juice agar plates overnight at 25°C. Embryos were dechorionated in bleach. Stage 15 embryos were used, unless stated otherwise. The appropriate genotype of embryos was identified based on the presence of fluorescent probes and/or the

absence of balancer chromosome expressing fluorescent markers. The genotypes of the embryos used in each experiment are listed in the next section.

## Genotypes used in each experiment
Figure 1
a, b, c, d
*w; Sn-Gal4, UAS-LifeActGFP*
e
"Moesin[WT]": *w;; Srp-Gal4 / UAS-Moesin[WT]*
"PIP₂": *w;; Srp-Gal4 / UAS-PLC-PH-GFP*
f
"MPAct-F-actin": *w;; Sn-Gal4, UAS-MPAct-mNeonGreen, UAS-F-tractin-mScarlet*
"MPAct-Plasma membrane": *w;; Sn-Gal4, UAS-MPAct-mNeonGreen, UAS-CaaX-mScarlet*

Figure 2
a, b, c, d
"Control": *w; Sn-Gal4, UAS-RedStinger, UAS-LifeActGFP*
"Moesin[Mut]": *Moesin[PL106] / y; Sn-Gal4, UAS-RedStinger, UAS-LifeActGFP*
e, f
"Control": *w;; Sn-Gal4, UAS-MPAct-mNeonGreen, UAS-CaaX-mScarlet*
"Moesin[Mut]": *Moesin[PL106] / y;; Sn-Gal4, UAS-MPAct-mNeonGreen, UAS-CaaX-mScarlet*

Figure 3
a, b, c
"Control": *w; Sn-Gal4, UAS-RedStinger, UAS-LifeActGFP*
"Moesin[Mut]": *Moesin[PL106] / y; Sn-Gal4, UAS-RedStinger, UAS-LifeActGFP*
d
"Control": *w; Srp-Gal4, UAS-RedStinger, UAS-CLIP-GFP*
"Moesin[Mut]": *Moesin[PL106] /y; Srp-Gal4, UAS-RedStinger, UAS-CLIP-GFP*
f
"Moesin[Mut]": *Moesin[PL106] / y; Sn-Gal4, UAS-RedStinger, UAS-LifeActGFP*
g
"Control": *w; Sn-Gal4, UAS-RedStinger, UAS-LifeActGFP*
"Moe[Mut]": *Moesin[PL106] / y; Sn-Gal4, UAS-RedStinger, UAS-LifeActGFP*
"Moe RNAi": *w; UAS-Moesin RNAi (v37917); Sn-Gal4, UAS-RedStinger, UAS-LifeActGFP*
"pten[Mut]": *w; Pten[3]; Sn-Gal4, UAS-RedStinger, UAS-LifeActGFP*
"sktl[Mut]": *w; sktl[Δ20]; Sn-Gal4, UAS-RedStinger, UAS-LifeActGFP*
"slik[Mut]": *w; Slik[KG08374]; Sn-Gal4, UAS-RedStinger, UAS-LifeActGFP*
"slik RNAi": *w; Sn-Gal4, UAS-RedStinger, UAS-LifeActGFP; UAS-Slik RNAi*

Figure 4
c
"Moesin[WT]": *w;; Sn-Gal4, UAS-RedStinger, UAS-Moesin[WT]-GFP*
"Moesin[KN]": *w;; Sn-Gal4, UAS-RedStinger, UAS-Moesin[KN]-GFP*
"Moesin[TD]": *w; Sn-Gal4, UAS-RedStinger, UAS-Moesin[TD]-GFP*
"Moesin[TDKN]": *w; Sn-Gal4, UAS-RedStinger, UAS-Moesin[TDKN]-GFP*
"Moesin[TA]": *w; Sn-Gal4, UAS-RedStinger, UAS-Moesin[TA]-GFP*
"Moesin[TAKN]": *w; Sn-Gal4, UAS-RedStinger, UAS-Moesin[TAKN]-GFP*
d,e,f,g,h
"Moesin[WT]": *w; UAS-Moesin[WT]; Sn-Gal4, UAS-MPAct-mNeonGreen, UAS-CaaX-mScarlet*
"Moesin[TD]": *w; UAS-Moesin[TD]; Sn-Gal4, UAS-MPAct-mNeonGreen, UAS-CaaX-mScarlet*

Figure 5
a,b,c
"Moesin[WT]": *w;; Srp-Gal4 / UAS-Moesin[WT]-GFP*
"Moesin[KN]": *w;; Srp-Gal4 / UAS-Moesin[KN]-GFP*
"Moesin[TD]": *w; UAS-Moesin[TD]-GFP / +; Srp-Gal4 / +*

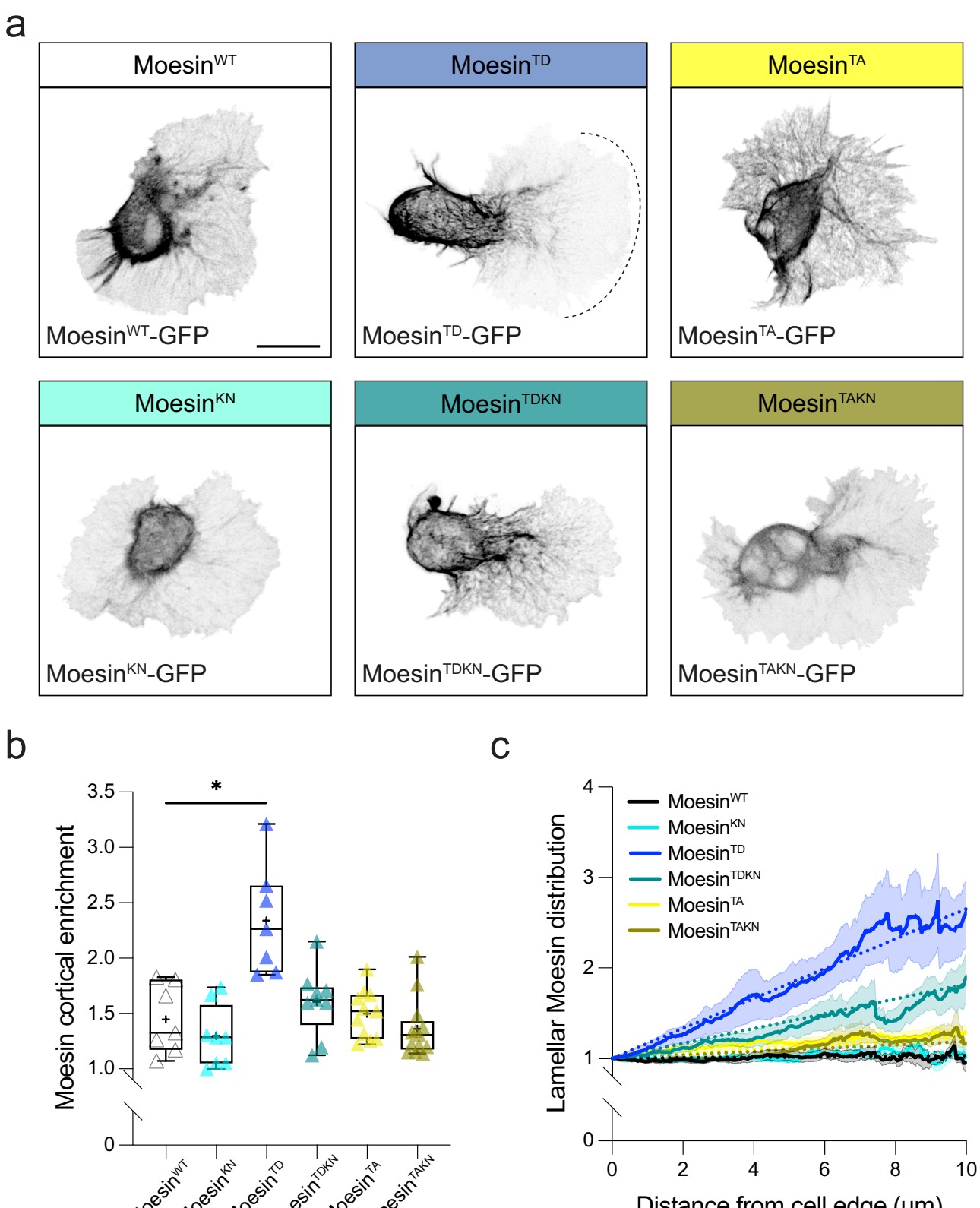

**Fig. 5 | Moesin distribution is affected by a combination of phosphorylation and PIP₂ binding. a** Imaging of hemocytes expressing single copies of GFP-tagged Moesin constructs. **b** Quantification of GFP-tagged Moesin transgenes highlighting an enhanced cortical enrichment of Moesin$^{TD}$, which is reduced when combined with the K/N mutation. *$P$ = 0.0134, Kruskal–Wallis test and Dunn's multiple comparison test. Boxplot shows medians, 25th and 75th percentiles as box limits, minimum and maximum values as whiskers; each datapoint is displayed as a marker

($n$ = 7 Moesin$^{WT}$, 8 Moesin$^{KN}$, 8 Moesin$^{TD}$, 9 Moesin$^{TDKN}$, 11 Moesin$^{TA}$ and 13 Moesin$^{TAKN}$ hemocytes). **c** Line scan analysis of GFP-tagged Moesin constructs within the hemocyte lamella normalized from the leading edge of the cell high-lighting an enhanced Moesin$^{TD}$ gradient, which is reduced when combined with the K/N mutation; data are presented as mean values +/- standard error of the mean ($n$ = 7 Moesin$^{WT}$, 8 Moesin$^{KN}$, 7 Moesin$^{TD}$, 10 Moesin$^{TDKN}$, 11 Moesin$^{TA}$ and 13 Moesin$^{TAKN}$ hemocytes). Scale bars, 10 μm.

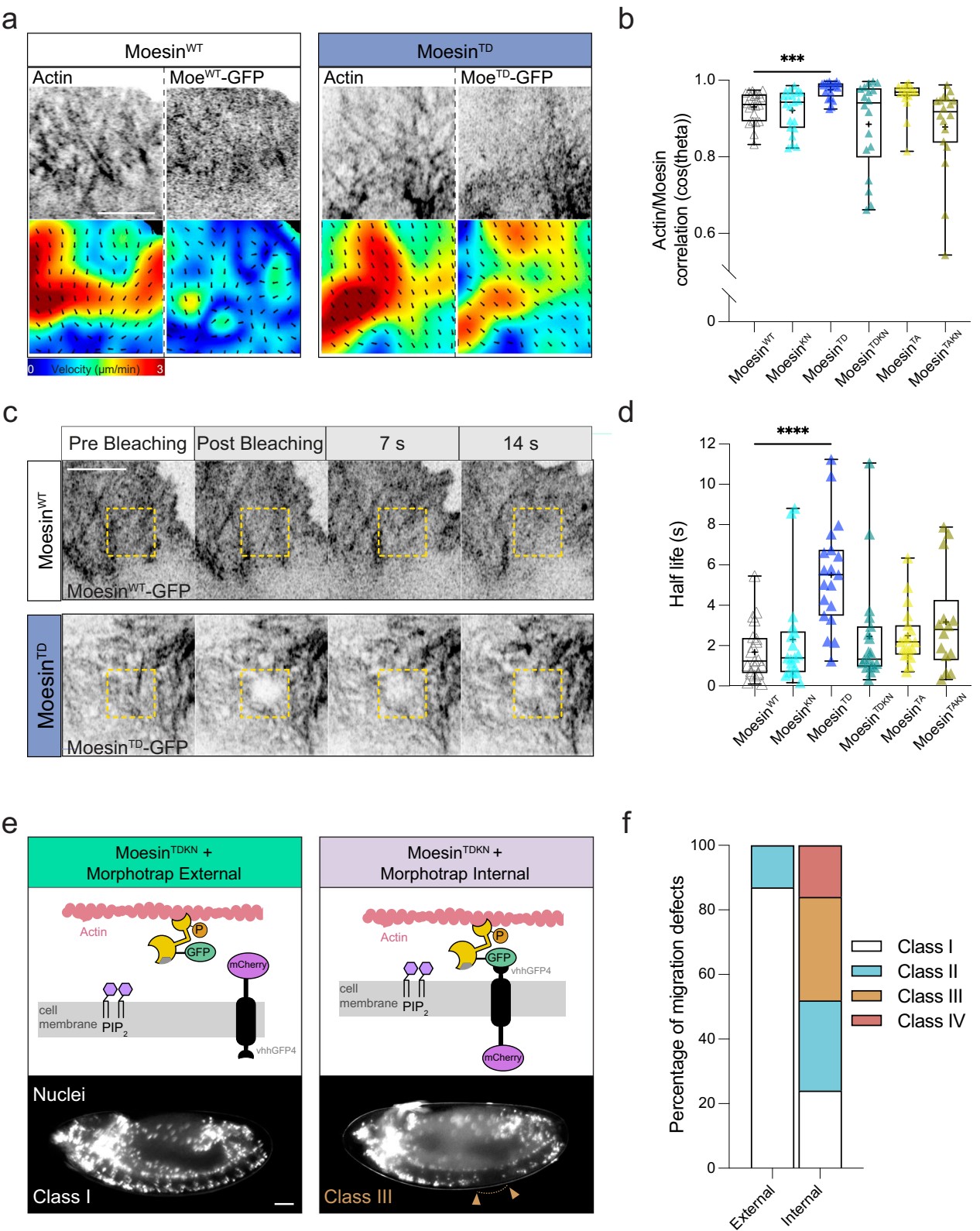

"Moesin$^{TDKN}$": w; UAS-Moesin$^{TDKN}$-GFP / +; Srp-Gal4 / +
"Moesin$^{TA}$": w; UAS-Moesin$^{TA}$-GFP / +; Srp-Gal4 / +
"Moesin$^{TAKN}$": w; UAS-Moesin$^{TAKN}$-GFP / +; Srp-Gal4 / +
Figure 6
a
"Moesin$^{WT}$": w;Srp3X-MoeCherry / +; Srp-Gal4 / UAS-Moesin$^{WT}$ -GFP
"Moesin$^{TD}$": w; UAS-Moesin$^{TD}$-GFP / Srp3X-MoeCherry; Srp-Gal4 / +

b
"Moesin$^{WT}$": w;Srp3X-MoeCherry / +; Srp-Gal4 / UAS-Moesin$^{WT}$ -GFP
"Moesin$^{KN}$": w; Srp3X-MoeCherry / +; Srp-Gal4 / UAS-Moesin$^{KN}$-GFP
"Moesin$^{TD}$": w; UAS-Moesin$^{TD}$-GFP / Srp3X-MoeCherry; Srp-Gal4 / +
"Moesin$^{TDKN}$": w; UAS-Moesin$^{TDKN}$-GFP / Srp3X-MoeCherry; Srp-Gal4 / +
"Moesin$^{TA}$": w; UAS-Moesin$^{TA}$-GFP / Srp3X-MoeCherry; Srp-Gal4 / +

**Fig. 6 | Moesin phosphorylation and PIP$_2$ binding are required for Moesin advection by actin retrograde flow to regulate hemocyte motility. a** Live imaging and particle image velocimetry (PIV) analysis of actin and Moesin$^{WT}$ or Moesin$^{TD}$ within hemocyte lamellae revealing coordination of actin and Moesin$^{TD}$ flows. Scale bar, 5 μm. **b** Quantification of the correlation of motion (cos(theta)) between actin and Moesin transgenes revealing an increased coordination between actin and Moesin$^{TD}$, which is reduced when combined with the K/N mutation. ***$P$ = 0.0008, Kruskal–Wallis test and Dunn's multiple comparison test. ($n$ = 22 Moesin$^{WT}$, 21 Moesin$^{KN}$, 20 Moesin$^{TD}$, 18 Moesin$^{TDKN}$, 19 Moesin$^{TA}$ and 18 Moesin$^{TAKN}$ hemocytes. **c** Fluorescence recovery after photobleaching (FRAP) analysis of GFP-tagged Moesin$^{WT}$ or Moesin$^{TD}$ transgenes within hemocyte lamella (dashed square). Scale bar, 5 μm. **d** Quantification of the half life of fluorescence recovery revealing reduced diffusivity of Moesin$^{TD}$, which is restored when combined with the K/N

mutation. ****$P$ < 0.0001, Kruskal–Wallis test and Dunn's multiple comparison test. ($n$ = 22 Moesin$^{WT}$, 20 Moesin$^{KN}$, 20 Moesin$^{TD}$, 19 Moesin$^{TDKN}$, 19 Moesin$^{TA}$ and 17 Moesin$^{TAKN}$ hemocytes). **e** (top panels) Schematic highlighting the Morphotrap strategy to drive localization of GFP-tagged Moesin transgenes to the outer or inner face of the plasma membrane. (bottom panels) Example images of hemocyte dispersal phenotypes when Moesin$^{TDKN}$ is recruited to the external or internal face of the plasma membrane. Scale bar, 50 μm. **f** Quantification of hemocyte dispersal defects when Moesin$^{TDKN}$ is recruited to the plasma membrane with Morphotraps highlighting that artificially recruiting Moesin$^{TDKN}$ to the internal face of the plasma membrane induces hemocyte dispersal defects ($n$ = 23 Morphotrap external and 25 Morphotrap internal embryos). All boxplots show medians, 25th and 75th percentiles as box limits, minimum and maximum values as whiskers; each datapoint is displayed as a marker. Scale bars, 10 μm.

---

*"Moesin$^{TAKN}$": w; UAS-Moesin$^{TAKN}$-GFP / Srp3X-MoeCherry; Srp-Gal4 / +*

c

*"Moesin$^{WT}$": w;Srp3X-MoeCherry / +; Srp-Gal4 / UAS-Moesin$^{WT}$ -GFP*
*"Moesin$^{TD}$": w; UAS-Moesin$^{TD}$-GFP / Srp3X-MoeCherry; Srp-Gal4 / +*

d

*"Moesin$^{WT}$": w;Srp3X-MoeCherry / +; Srp-Gal4 / UAS-Moesin$^{WT}$ -GFP*
*"Moesin$^{KN}$": w; Srp3X-MoeCherry / +; Srp-Gal4 / UAS-Moesin$^{KN}$-GFP*
*"Moesin$^{TD}$": w; UAS-Moesin$^{TD}$-GFP / Srp3X-MoeCherry; Srp-Gal4 / +*
*"Moesin$^{TDKN}$": w; UAS-Moesin$^{TDKN}$-GFP / Srp3X-MoeCherry; Srp-Gal4 / +*
*"Moesin$^{TA}$": w; UAS-Moesin$^{TA}$-GFP / Srp3X-MoeCherry; Srp-Gal4 / +*
*"Moesin$^{TAKN}$": w; UAS-Moesin$^{TAKN}$-GFP / Srp3X-MoeCherry; Srp-Gal4 / +*

e

*"Moesin$^{TDKN}$ Morphotrap External": w; Sn-Gal4, UAS-RedStinger, UAS-Moesin$^{TDKN}$-GFP; UAS-Morphotrap External / +*
*"Moesin$^{TDKN}$ Morphotrap Internal": w; Sn-Gal4, UAS-RedStinger, UAS-Moesin$^{TDKN}$-GFP; UAS-Morphotrap Internal / +*

Figure 7

a,b,c,d,e,g,h

*"Control": w; Sn-Gal4, UAS-RedStinger, UAS-LifeActGFP*
*"Moesin$^{Mut}$": Moesin$^{PL106}$ / y; Sn-Gal4, UAS-RedStinger, UAS-LifeActGFP*

i

*"Moesin$^{WT}$": w; UAS-Moesin$^{WT}$; Sn-Gal4, UAS-RedStinger, UAS-GMA-GFP*
*"Moesin$^{TD}$": w; UAS-Moesin$^{TD}$; Sn-Gal4, UAS-RedStinger, UAS-GMA-GFP*

j

*"Moesin$^{WT}$": w; UAS-Moesin$^{WT}$; Sn-Gal4, UAS-RedStinger, UAS-GMA-GFP*
*"Moesin$^{KN}$": w; UAS-Moesin$^{KN}$; Sn-Gal4, UAS-RedStinger, UAS-GMA-GFP*
*"Moesin$^{TD}$": w; UAS-Moesin$^{TD}$; Sn-Gal4, UAS-RedStinger, UAS-GMA-GFP*
*"Moesin$^{TDKN}$": w; UAS-Moesin$^{TDKN}$; Sn-Gal4, UAS-RedStinger, UAS-GMA-GFP*
*"Moesin$^{TA}$": w; UAS-Moesin$^{TA}$; Sn-Gal4, UAS-RedStinger, UAS-GMA-GFP*
*"Moesin$^{TAKN}$": w; UAS-Moesin$^{TAKN}$; Sn-Gal4, UAS-RedStinger, UAS-GMA-GFP*

Supplementary Fig. 1

a,b,d

w;; Tub-Gal4 / UAS-MPAct-mNeonGreen, UAS-CaaX-mScarlet

c

w;; Sn-Gal4, UAS-MPAct-mNeonGreen, UAS-CaaX-mScarlet

Supplementary Fig. 2

a,b

*"Moesin$^{WT}$": w;; Sn-Gal4, UAS-RedStinger, UAS-Moesin$^{WT}$ -GFP / +*
*"Moesin$^{KN}$": w;;; Sn-Gal4, UAS-RedStinger, UAS-Moesin$^{KN}$-GFP / +*
*"Moesin$^{TD}$": w; Sn-Gal4, UAS-RedStinger, UAS-Moesin$^{TD}$-GFP / +*
*"Moesin$^{TDKN}$": w; Sn-Gal4, UAS-RedStinger, UAS-Moesin$^{TDKN}$-GFP / +*
*"Moesin$^{TA}$": w; Sn-Gal4, UAS-RedStinger, UAS-Moesin$^{TA}$-GFP / +*
*"Moesin$^{TAKN}$": w; Sn-Gal4, UAS-RedStinger, UAS-Moesin$^{TAKN}$-GFP / +*

Supplementary Fig. 3

a,b

*"Control": w; Sn-Gal4, UAS-Zip-GFP*
*"Moesin$^{Mut}$": Moesin$^{PL106}$ / y; Sn-Gal4, UAS-Zip-GFP*

c

*"Cell1 and Cell2": w; Sn-Gal4, UAS-Moe$^{TD}$-GFP / UAS-Sqh-mCherry*

d

*"Myosin$^{Mut}$": w; Zip$^1$/Zip$^1$; Sn-Gal4, UAS-LifeAct-GFP; dataset reused from our previous work[14]*
*"Moesin$^{Mut}$": Moesin$^{PL106}$ / y; Sn-Gal4, UAS-LifeAct-GFP*

Supplementary Movie 1, 2, and 3

w; Sn-Gal4, UAS-LifeActGFP

Supplementary Movie 4

*"Moesin$^{WT}$": w;; Srp-Gal4 / UAS-Moesin$^{WT}$*
*"PIP$_2$": w;; Srp-Gal4 / UAS-PLC-PH-GFP*

Supplementary Movie 5 and 6

*"Top panel": w;; Sn-Gal4, UAS-MPAct-mNeonGreen, UAS-F-tractin-mScarlet*
*"Bottom panel": w;; Sn-Gal4, UAS-MPAct-mNeonGreen, UAS-CaaX-mScarlet*

Supplementary Movie 7

*"Control": w;; Sn-Gal4, UAS-MPAct-mNeonGreen, UAS-CaaX-mScarlet*
*"Moesin$^{Mut}$": Moesin$^{PL106}$ / y;; Sn-Gal4, UAS-MPAct-mNeonGreen, UAS-CaaX-mScarlet*

Supplementary Movie 8

*"Control": w; Sn-Gal4, UAS-RedStinger, UAS-LifeActGFP*
*"Moesin$^{Mut}$": Moesin$^{PL106}$ / y; Sn-Gal4, UAS-RedStinger, UAS-LifeActGFP*

Supplementary Movie 9

*"Moesin$^{WT}$": w; UAS-Moesin$^{WT}$; Sn-Gal4, UAS-MPAct-mNeonGreen, UAS-CaaX-mScarlet*
*"Moesin$^{TD}$": w; UAS-Moesin$^{TD}$; Sn-Gal4, UAS-MPAct-mNeonGreen, UAS-CaaX-mScarlet*

Supplementary Movie 10

w; UAS-Moesin$^{TD}$-GFP / Srp3X-MoeCherry; Srp-Gal4 / +

Supplementary Movie 11

w; Sn-Gal4, UAS-LifeActGFP

## Methods

### Constructions of UAS-CaaX, F-tractin, MPAct and Sqh

**pUASt-attB-CaaX-mScarlet.** was generated by inserting an 808 bp fragment (synthetized by BioBasic) containing mScarlet-I[53] upstream

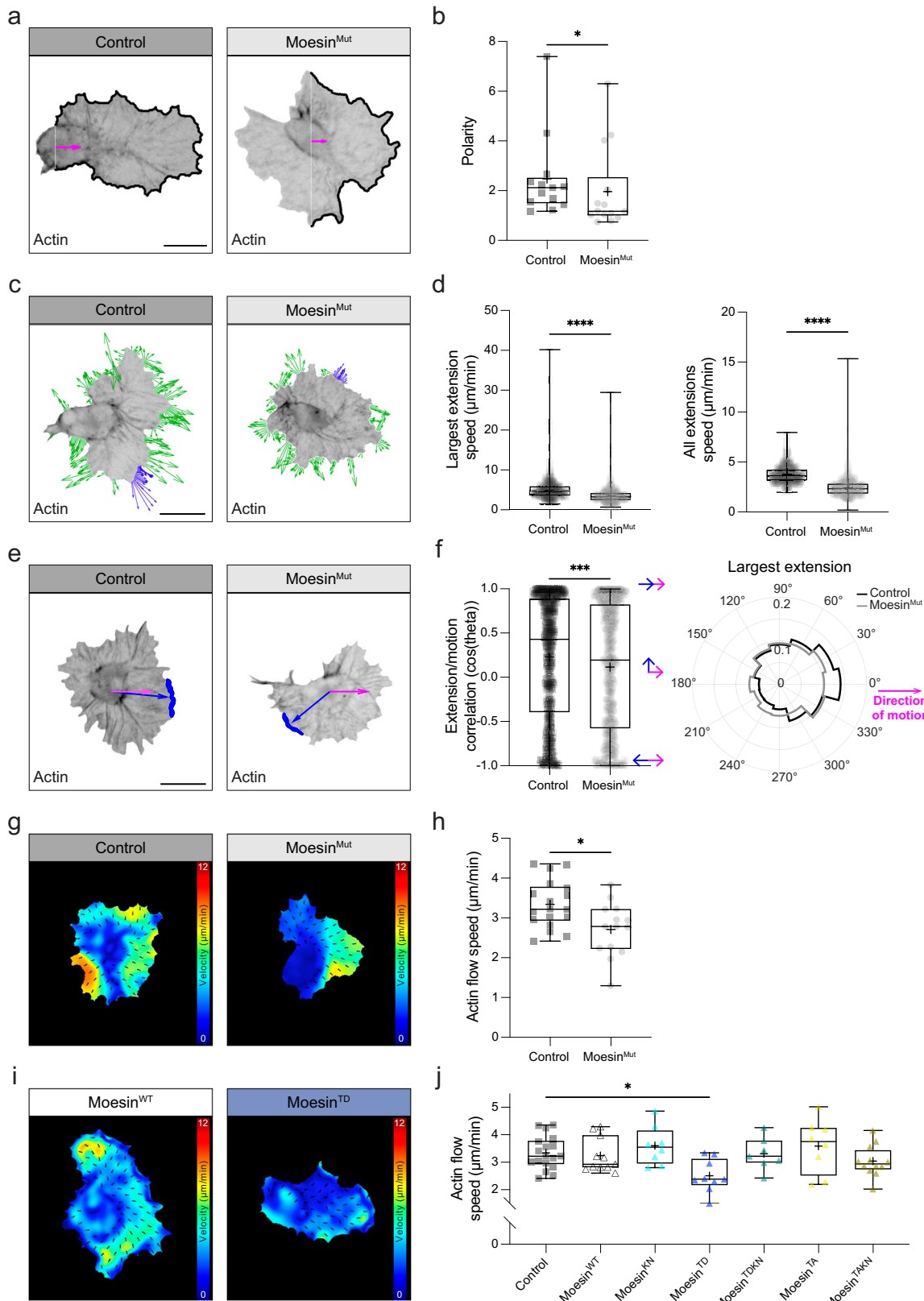

of the CaaX membrane-targeting motif of the *Drosophila* Ras2 sequence (from ref.[54]) into the linearized PacI-AvrII pUASt-attB plasmid, using ligation T4 strategy (New England Biolabs, Inc.).

The construct was sequenced (Eurofins Genomics) using the following sequencing primers:

5′GGAATTGGGAATTCGTTTTAATTAAATGGTGAGCAAGGGCGAG3′
5′GATCCTCTAGAGGTACCTAGGCTACATCAGGCAGCACTTCCT3′

**pUASt-attB-F-tractin-mScarlet.** was generated by inserting an 877 bp fragment (synthetized by BioBasic) containing F-tractin (from[55])

**Fig. 7 | Moesin is required to regulate hemocyte polarity, leading edge dynamics, and actin retrograde flow. a** Morphodynamic analysis of hemocyte polarity. **b** Quantification of polarity highlighting a reduction in the absence of Moesin. *P = 0.0332, Mann–Whitney two-tailed test. (*n* = 13 Control and 14 Moesin^Mut hemocytes). **c** Dynamic analysis of leading edge velocities highlighting the edge velocity of all extensions (green) or the largest contiguous extension (blue). Extension vector magnitude has been multiplied by 10 for visualization purposes. **d** Quantification of speed of the largest extension or all extensions highlighting a reduction in the absence of Moesin. ****P < 0.0001, Mann–Whitney two-tailed test. (*n* = 677 Control and 995 Moesin^Mut timepoints for 13 and 14 hemocytes, respectively). **e** Comparison of the direction of cell motion (magenta) with the direction of the largest extension (blue) in Control and Moesin^Mut hemocytes. **f** (left panel) Quantification of the correlation in motion (cos(theta)) between the largest extension (blue) and the direction of movement (magenta) in Control and Moesin^Mut hemocytes highlighting a decreased correlation in the absence of Moesin. (right panel) Rose plot comparing the direction of the largest extension

normalized to the instantaneous hemocyte direction of motion. ***P = 0.0008, Mann–Whitney two-tailed test. (*n* = 677 Control and 995 Moesin^Mut timepoints for 13 and 14 hemocytes, respectively). **g** PIV analysis of actin flow in Control and Moesin^Mut hemocytes. **h** Quantification of actin flow speed in the absence of Moesin. *P = 0.0128, Mann–Whitney two-tailed test. (*n* = 18 Control and 14 Moesin^Mut hemocytes). (n = 18 Control and 14 Moesin^Mut hemocytes). **i** PIV analysis of actin flow when expressing two copies of Moesin^WT or Moesin^TD. **j** Quantification of actin flow speed when expressing two copies of Moesin transgenes revealing a reduction in retrograde flow in Moesin^TD cells, which is rescued when combining with a K/N mutation. *P = 0.0286, Kruskal–Wallis test and Dunn's multiple comparison test. (*n* = 18 Control, 11 Moesin^WT, 8 Moesin^KN, 9 Moesin^TD, 7 Moesin^TDKN, 8 Moesin^TA and 12 Moesin^TAKN hemocytes). All boxplots show medians, 25th and 75th percentiles as box limits, minimum and maximum values as whiskers; each datapoint is displayed as a marker. Scale bars, 10 μm.

---

downstream of the mScarlet-I sequence[53] into the linearized PacI-AvrII pUASt-attB plasmid, using ligation T4 strategy (New England Biolabs, Inc.).

The construct was sequenced (Eurofins Genomics) using the following sequencing primers:

5′GGAATTGGGAATTCGTTTTAATTAAATGGGCATGGCCCGC3′
5′GATCCTCTAGAGGTACCTAGGCTACTTGTACAGCTCGTCCATGC3′

**pUASt-attB-MPAct-mNeonGreen.** was generated by inserting a 952 bp fragment (synthetized by BioBasic) containing F-tractin (from ref. [55]) downstream of the mNeonGreen sequence, and upstream of the CaaX membrane-targeting motif of the *Drosophila* Ras2 sequence (from ref. [54]) into the linearized PacI-AvrII pUASt-attB plasmid, using ligation T4 strategy (New England Biolabs, Inc.).

The construct was sequenced (Eurofins Genomics) using the following sequencing primers:

5′GGAATTGGGAATTCGTTTTAATTAAATGGGCATGGCCCGC 3′
5′GATCCTCTAGAGGTACCTAGGCTACATCAGGCAGCACTTCCT 3′

**pUASt-attB-Sqh-mCherry.** was generated by inserting an 1254 bp fragment containing mCherry[56] downstream of the spaghetti squash (Sqh) complete coding sequence (without stop codon) into the linearized EcoRi/XhoI pUASt-attB plasmid, using ligation T4 strategy (synthetized by Thermo Fisher Scientific).

The plasmids containing mScarlet were injected into flies (BDSC 9744) harboring an attP-9A insertion at 3 R chromosome (89E11), the plasmid containing mNeonGreen was injected into flies (BDSC 9725) harboring an attP-9A insertion at 3 L chromosome, the plasmid containing Sqh-mCherry was injected into flies (BDSC 8622) harboring an attP-2 insertion at 3 L chromosome, injections made by BestGene.

### Sample preparation and mounting for imaging

For all the embryo imaging, dechorionated embryos were mounted in 10S Voltalef oil (VWR) between a glass coverslip covered with heptane glue and a gas-permeable Lumox culture dish (Sarstedt) as described previously in ref. [15]. For the salivary gland imaging, stage L3 larvae were dissected to isolate the tissue, which was mounted in PBS between a slide and glass coverslip as described previously in ref. [19]. For the gut and ovaries imaging, adult young females were dissected to isolate the tissues, which were mounted in PBS between a slide and glass coverslip as described previously in refs. [17,18,21].

### Widefield and confocal microscopy

Widefield images were acquired using an M205 fluorescent dissection microscopy (Leica) equipped with a PLANAPO 2.0x objective. Confocal images were acquired with a LSM880 confocal microscope (Carl Zeiss) equipped with a 63x NA 1.4 Apochromat oil objective with

a zoom of 1, unless stated otherwise. For border cells imaging, a zoom of 2 was used. The Super Resolution Airyscan mode with a zoom of 1.8 was used for imaging of the digestive tract, and for the follicular epithelium image a zoom of 2 was used. For high-resolution images of single hemocytes, the Super Resolution Airyscan mode was used with a zoom of 3. FRAP experiments were performed with the Super Resolution Airyscan mode using a zoom of 5 in the lamella, and only a region of interest of 266 × 283-pixel acquired. After taking 5 images, regions of interest (with a diameter of 16 pixels) were bleached twice with a 405 nm, 458 nm, 488 nm and 561 nm lasers at 100% laser transmission with 106.28 μs/pixel dwell time, immediately followed by acquisition of 45 series of images every 0.5 s. For high-resolution live imaging, the Super Resolution Airyscan mode was used with a zoom of 1.4 for the MPAct embryos, and 1.8 for the Moesin^TD embryo. For live imaging of the whole embryo, a 40x NA 1.3 Apochrimat oil objective with a zoom of 0.9 (time-lapse every 20 s) was used. A PerkinElmer Ultraview spinning disk equipped with a ×63 NA 1.4 Plan-Apochromat oil objective was used to study the lamella actin network (time-lapse every 5 s) and the contact repulsion (time-lapse every 10 s)[15].

### Cell volume and shape measurements

To visualize and measure cell volume and shape, cells were labelled with UAS-LifeAct-GFP or UAS-CaaX-mScarlet under the expression of hemocyte driver sn-Gal4. The cell body of each cell was manually segmented across the whole z-projection in Fiji. Segmented cell bodies were opened in Imaris and the "Surface" function was used to calculate volume and sphericity. A similar protocol was used to calculate volume and sphericity of whole cells. Lamellae volumes were calculated by subtracting cell body volumes from whole cell volumes.

### MPAct ratio analysis

MPAct ratios were generated as previously described by Bisaria et al. 2020[12], taking the ratio of the MPAct/CaaX or MPAct/F-tractin channels. The mean of ratiometric MPAct over CaaX intensities (here termed MPAct/CaaX or MPAct ratio) was used to measure the actin cortex in the different genotypes. The mean was calculated from segmented hemocyte images obtained from max intensity projection of super resolution images.

### Fluorescence recovery after photobleaching (FRAP) analysis

Moesin diffusion in the lamella was studied using the max intensity projection of super resolution images. The FRAP analysis was calculated as previously described[57]. Using Fiji, three regions of interest (ROIs) were drawn, a 70 × 70-pixel ROI on top of the bleaching area (ROI^Bleach), a 70 × 70-pixel ROI on top of a control area (ROI^control), and a 20 × 20-pixel ROI on the background area (ROI^BG). The pre-bleaching

signal is the average signal of the first five frames before bleaching. Time 0 is the first time acquired after the bleaching. The Eq. 1 was used to apply the photobleaching correction to each time point, after that, the Eq. 2 was used to normalize the values. Values from time 0 to 14 s were analyzed in Prism, to calculate a nonlinear regression curve to obtain the recovery half life. Half life values for each image were plotted per genotype.

$$Value_{t(n)}^{Corrected} = \left( ROI_{t(n)}^{Bleach} - ROI_{t(n)}^{BG} \right) * \left( \frac{ROI_{t(pre-bleach)}^{Control}}{ROI_{t(n)}^{Control}} \right) \quad (1)$$

$$Value_{t(n)}^{Normalised} = \frac{\left( Value_{t(n)}^{Corrected} - Value_{t(0)}^{Corrected} \right)}{\left( Value_{t(pre-bleach)}^{Corrected} - Value_{t(0)}^{Corrected} \right)} \quad (2)$$

### Analysis of hemocyte migration in the four classes of dispersal defects

Images of lateral views of stage 13 embryos of the adequate genotype were selected, imaged and analyzed as described in refs. 58,59. Embryos were classified in four distinct phenotypic classes according to the number of neuromeres devoid of hemocytes in the ventral nerve cord and the percentage of each class was calculated per genotype.

### Hemocyte speed

To calculate the average hemocyte speed for each embryo, up to 94 hemocyte nuclei were tracked using the Imaris "Plot" function.

### Particle image velocimetry (PIV) analysis

PIV analysis was performed to quantify actin and Moesin flows within the lamellae of randomly migrating hemocytes. Hemocyte lamellae are first manually segmented from the background using Fiji. Subsequently, PIV analysis is carried out as described in previous work[15] with an open-source MATLAB suite that can be found at this link: https://github.com/stemarcotti/PIV. Each frame is first divided into small windows ($1.2 \, \mu m^2$, with $0.8 \, \mu m$ overlap); each window is then searched for in the subsequent frame within an area of $2 \, \mu m^2$. If a correlation score between the source and the search windows higher than 0.5 is achieved, the match is considered a successful tracking event, and a vector is computed to obtain a velocity vector field. The vector field is then interpolated with a Gaussian kernel in space (size $5 \, \mu m$, sigma $1 \, \mu m$) and in time (size 5 frames, sigma 2 frames).

All generated colormaps highlight the magnitude of the velocity, vectorial representation specifies the direction of the flow field. Alternatively, the vector field can be represented as streamlines, which are lines tangent to each PIV vector, highlighting the path that a massless object would take if dropped in the flow field.

The PIV vector field was averaged in space and time across the full length of each movie to obtain the average velocity measurements presented. The flow vector field was used to create the deformation map, by computing the accumulated deformations on the vertices of a regular grid. The cosine similarity between all vectors in the PIV field was calculated when comparing the actin and Moesin flows for directional correlation.

### Divergence

Negative divergence measures the presence of sinks within the flow field, and it highlights regions where the actin network is compressed[14]. Divergence was quantified in MATLAB (MathWorks, version 2023a) with a central difference scheme to compute the spatial derivatives of the flow velocity field obtained by PIV. The cortical region was defined as a $5 \, \mu m$ ring around the cell body.

### Lamellar distribution

Images of segmented hemocytes were used to calculate in MATLAB the Moesin or MPAct ratio distribution across the lamellae. First, we manually masked the cell body and the regions of the lamellae where no large actin fibers were present. Next, we computed an ellipse centered at the cell body centroid circumscribing the cell, and we drew line scans with a thickness of 10 px every 100 px along the ellipse edge. These line scans were used to interrogate signal intensity on the masked regions of the lamellae, aligned to start at the cell edge towards the cell body. An average profile per cell was obtained and plotted for comparison between samples.

### Cortical enrichment

Cortical enrichment measured whether a protein of interest is more localized in the cortical region compared to the cell body. To compute this metric, we used a similar approach described above for the lamellar distribution, by drawing 10 px-thick line scans from the cell body centroid to the edge of the lamellae. The line scans were then aligned by using an ellipse circumscribing the cell body as a reference (set as x = 0) and averaged across the cell for every location where more than 10% of data was available (both for negative and positive values of x, representing the cell body and the lamella respectively). The cortical enrichment was defined as the ratio between the signal intensity at the cell body edge (x = 0) vs. at the cell body center (set to x = -3).

### Persistence

Tracking of nuclei of randomly migrating hemocytes was performed in MATLAB by finding the centroid of the nuclear binary mask at every time point. Persistence was calculated in a walking average manner on 60 s intervals as the distance between the start and end point divided by the total track length within each interval. This metric takes values of 1 when the object moves in a straight line (start-end distance equals track length), and lower values the more erratic the motion.

### Contact inhibition of locomotion (CIL)

CIL analysis was performed as previously described[15,50,60] in MATLAB. The time of the collision (t = 0 s) was defined by visually marking the microtubule alignment between two single migrating hemocytes coming into contact. The location of the two nuclei was then recorded 60 s before collision, at the time of collision, and 60 s after collision. The collision axis between two colliding partners was subsequently normalized to the horizontal cartesian axis, and we computed the acceleration coincident with the time of collision. The x-component of the acceleration was used as a metric as to whether the analyzed hemocyte reacted to the collision by repolarizing its migration away from the colliding partner.

### Morphodynamic analysis of cell polarity

Lamellar polarization was quantified by evaluating the cell shape in relationship to the direction of motion with a custom MATLAB code. At each time point, a binary mask representing the segmented migrating hemocyte was halved with a line perpendicular to the direction of nuclear motion passing through the nuclear centroid. Lamellar polarity was defined as the ratio between the front (area of the half towards the direction of motion) and the back of the cell (area in the half opposite to the direction of motion) averaged for each cell over the length of a movie.

### Cell edge analysis

Cell edge dynamics evaluation in randomly migrating hemocytes was carried out as previously described in ref. 14. Edge extensions at

each pixel on the cell boundary were computed using the Segmentation and Windowing package[61] and we calculated in MATLAB at every time point the speed of all extensions and the largest extension (defined as the longest uninterrupted section along the hemocyte perimeter).

We computed two vectors from the nuclear center at each time point: one representing the cell direction of motion (pointing towards the nuclear center at the subsequent time point), and one highlighting the direction to the largest extension (pointing towards the region with the largest area calculated within the difference between the cell mask at the subsequent and current time points). Cosine similarity between these two vectors was computed to evaluate angle correlation, and rose plots were generated by normalizing the direction of motion to the horizontal cartesian axis.

### Statistics and reproducibility

Statistical tests and n-numbers are reported in figure captions; statistical analysis was performed in GraphPad Prism (version 10). Significance is indicated as follows: '****' for $p$-values lower than 0.0001, '***' lower than 0.001, '**' lower than 0.01, '*' lower than 0.05, 'ns' otherwise. A minimum of ten cells were analyzed when performing qualitative assessments.

### Reporting summary

Further information on research design is available in the Nature Portfolio Reporting Summary linked to this article.

## Data availability

Source data for figures are provided with this paper, all other data supporting the findings of this study are available from the corresponding author upon reasonable request. Source data are provided with this paper.

## Code availability

The computational analysis was performed in MATLAB (MathWorks, version R2023a) using custom code, which is available on GitHub or can be obtained from the corresponding author upon reasonable request.

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

## Acknowledgements

This project has been funded by the Wellcome Trust (grant no. 107859/Z/15/Z to B.M.S., grant no. 22460/Z/21/Z to W.W., and grant no. 107355/Z/15/Z to A.J.D.), the European Research Council (ERC) under the European Union's Horizon 2020 research and innovation programme (grant agreement no. 681808 to B.M.S.), the Medical Research Council (grant no. MR/W017407/1 to B.M.S.), and the Biotechnology and Biological Sciences Research Council (grant no. BB/V006169/1 to B.M.S.). For the purpose of open access, the author has applied a CC BY public copyright license to any Author Accepted Manuscript version arising from this submission. We thank Jennifer Zanet for *Drosophila* reagents.

## Author contributions

B.J.S.-S and B.M.S. conceived and designed the study, and provided guidance; B.J.S.-S., D.S.-G., M.-d.-C.D.-d.-l.-L. and M.B. performed the *Drosophila* experiments; B.J.S.-S., S.M., D.S.-G. and M.-d.-C.D.-d.-l.-L. analyzed the data; A.J.D. and W.W., resources; funding acquisition, A.J.D., W.W and B.M.S.; writing – original draft, B.M.S.; writing – review & editing, B.J.S.-S., S.M., D.S.-G., M.-d.-C.D.-d.-l.-L. and B.M.S.

## Competing interests

The authors declare no competing interests.
