## [Transparent Peer Review file · Nature Communications]

Moesin integrates cortical and lamellar actin networks during *Drosophila* macrophage migration

Corresponding Author: Professor Brian Stramer

Version 0:

Reviewer comments:

Reviewer #1

(Remarks to the Author)

This manuscript by Sanchez-Sanchez et al., investigate the role of the actin cortex in the migration of *Drosophila* hemocytes in vivo. Using powerful microscopy experiments and analysis and different mutants of Moesin, sole member of the Ezrin-Radixin-Moesin family in flies, they show that the actin cortex is mainly restricted to the cell body and do not extend in the lamella, and that perturbing the cortex impairs migration. In addition, they demonstrate that Moesin is a key regulator of this process and that its function requires both binding to the actin cytoskeleton and to the plasma membrane via PIP2. This manuscript is a very comprehensive study that is well suited for publication in Nature Communications. It will be of interest to a broad audience. Indeed, even if its findings are not extremely surprising, the manuscript demonstrates that Moesin has an impact on the actin dynamics and their work suggests that cortical Moesin impacts on lamellae functions.

I have only minor concerns that are listed below:

- Divergence in actin flow may not be clear for everyone (I needed a refresh). As such, the authors should explain the concept better. Also, they represent divergence on a scale of 0 to 7 or 0 to 12 in the figures which is different from their previous work (Yolland et al., 0 to -1). I suspect that the scale in the images should indeed be 0 to -1. If I am wrong and the scale is indeed correct, then the quantification provided (and the description in the text) makes little sense as it is in the negatives and does not compare to the respective images.

- In Fig.2, the authors found that lamellar size is unaffected, but the morphology of the lamella seems different in MoeMut (they seem shorter and thicker). It might be that this particular image is not representative of the phenotype, but, also when considering the results in Fig.7 (see below), a better characterisation of lamellae morphology to ensure that it is normal would reinforce the conclusions of the authors. This is particularly important as the authors assume for the rest of the paper that only cortical actin at the cell body is affected (again, see comments for Fig.7). The authors could for example determine the thickness, the length and the dynamics of the lamella in control and MoeMut.

- How does the authors explain that the MoeTAKN mutant expression lead to a phenotype as it is not associated with membrane nor actin ? Also that MoeTAKN seems cortical in Fig.5 ? This seems contradictory with their finding that both actin and membrane association are required for Moe accumulation at the cell body.

- Fig.6e the embryos are blurry (seems like a motion blur).

- l.189-190: I don't think that the Morphotrap external can recruit a protein that is intracellular...

- l.195: the conclusion that advection requires membrane binding could be reinforce by investigating the dynamics of the TDKN form with the Morphotrap internal.

- Fig.7 show that lamella is indeed affected when perturbing Moesin's activity. The authors conclude that the phenotype is basically due to the change at the cortex of the cell body. I am not sure about their argument there. Yes, most of Moe's role seems to be on the cell body, but there might be small amount of Moe in the lamella that might be sufficient to induce a phenotype. I am not sure how this could be tested (an optogenetic recruitment could work, but this seems to be excessively demanding at this stage), so if the authors cannot provide further data demonstrating that only cortical actin is perturbed, I

would suggest that the author tone down their conclusion.

- There is no "Acknowledgments" section.

Reviewer #2

(Remarks to the Author)

This beautiful study by Sanchez-Sanchez et al takes full advantage of the fly embryo and uses state of the art microscopy techniques to investigate the role of moesin in regulating the complex actin networks that drive macrophage migration in vivo. The imaging is exceptional and the authors use the powerful live imaging that the model system offers to characterise in unprecedented detail the organisation and dynamics of the actin network within fly macrophages (hemocytes) as they migrate within a developing fly. The authors show convincingly that these cells possess both a lamellar actin network at their front and a cortical actin network at the rear. Whilst the lamellar network has been previously studied the cortical network has not been looked at and consequently this study provides its first in depth characterisation which is an important piece of work. The fact that these cells present a hybrid actin organisation between a typically 'amoeboid' migratory cell and a 'mesenchymal' migratory cell is also fascinating. The authors show that the actin cortex regulator moesin plays an important role in regulating actin dynamics within the macrophages and intriguingly they show that both cortical and lamellipodial actin networks are inter-regulated. This is another fascinating observation and further work is required to work out the mechanism by which these two networks are coordinating their actions to drive migration.

I think to dissect out this mechanism would be beyond the scope of this work, however, in my opinion, focusing exclusively on these cells whilst they carry out developmental migration throughout the embryo misses an opportunity to expand the work to macrophage migration in a setting where they are carrying out a directed chemotactic migration to a focused source (for example a site of tissue injury or infection). Do the moesin mutants have a defect in this type of migration? In these settings how does the sensing of an external cue plug into the actin regulation that the authors observe in these cells to drive directed migration? In the majority of movies in the paper the cells are migrating randomly, how is moesin enabling the cell to polarise in response to an external signal (damage, infection, another cell to drive contact inhibition). If the authors can add data addressing this question to their study then I think it would have much wider appeal to the macrophage/immune cell migration community and that Nature Communication would be a perfect home for the paper. In addition, the following more minor points should be addressed:

1. Why is lamellar size unaffected in moesin mutants (Fig 2a,b) if moesin is required for the correct organisation of the cortical actin network and this network in turn regulates lamellipodial activity, would you not expect a difference in size as well as actin flow?
2. The phenotypes with moesin mutants, moeRNAi, pten and skt1 mutants are all stronger than that observed with slik mutants (figure 3g) Can the authors show further evidence for a role of slik in regulating moesin in the macrophages? Do they see genetic interaction between slik and moesin for example? If slik is not regulating moesin phosphorylation than are there other candidates?
3. Can the developmental migration defects be explained by a requirement for the actin cortex (and moesin) to allow the cell to withstand external forces coming from the spatial constraints experienced by macrophages as they carry out their migrations in the embryo?
4. This images of embryos in Figure 6e are out of focus and not up to the standard of the imaging throughout the rest of the paper. These need to be replaced.
5. The experiments using morphtrap internal and external are very elegant but beg the question - What is moesin binding to in the membrane? (i.e – what is the ligand depicted in Figure 4a within the migrating macrophages?). This may be beyond the scope of the current study but can the authors speculate?

Reviewer #3

(Remarks to the Author)

Sánchez-Sánchez et al. reported an interesting and important finding about how lamellar and cortical F-actin networks link and crosstalk with each other in *Drosophila* macrophage migration. In this process, active moesin binding to plasma membrane can be advected to the rear by lamellar actin flow to control cortical F-actin network. Then, cortical actin network can facilitate actin flow and leading protrusion dynamic growth. Finally, authors showed that this moesin-mediated crosstalk between lamellar and cortical F-actin networks are critical for macrophage migration. This study provided novel aspects of the relationship between two different F-actin networks and highlights the importance of active moesin in this process and cell migration. Overall, this manuscript is well introduced, well-organized and main findings are well supported by their experimental results. A couple of important points needs to be addressed before the publication of this story in Nat Commun: for example, whether and how advected active moesin controls myosin assembly in cortical F-actin network; whether cell migration is mainly dependent on leading protrusion dynamic growth, or cortical actomyosin contraction, or both. Answering these important questions will definitely improve the quality of this manuscript.

Major comments:

1. As previous study from this lab showed that myosin is also tightly related to actin flow in *Drosophila* macrophage (NCB 2019), it is unclear whether and how this advected/active moesin can affect myosin assembly at the cortical F-actin network. Authors should check myosin signals in the cortical region under some different genetic backgrounds.

2. If myosin assembly is really affected by advected/active moesin, authors might need to check whether myosin contraction might be able to influence macrophage migration. Regarding that myosin contraction might affect several different aspects, experimental answering of myosin function on cell migration might be difficult. Authors can discuss the potential effects of actomyosin contraction in cortical F-actin networks, on macrophage migration.

3. It is unclear how advected/active moesin controls macrophage migration, although authors stated that moesin can affect retrograde actin flow and leading protrusion dynamic growth. Authors can discuss these several possibilities to explain how cell migration can be controlled by moesin.

4. MoesinTD exhibits the most severe migration defects, and the authors analyzed the volume and also sphericity of cell body, and lamella, but did not show other phenotypes of others. Could authors analyze the same parameters of different mutants shown in Fig.4C?

Besides, authors showed a lot of quantification results of other two mutants, MoesinTA and MoesinTAKN, but authors did not mention the phenotypes/results of these two mutants (in figure 5, 6, 7) in the result texts.

5. Authors used 3D cell volume for quantification of cortical body. Is it possible that 2D area can also be used to present cell changes in cortical body? Is there some positive correlation between 3D cell volume and 2D cell area in cortical body? Authors should clarify this point.

Minor comments:

1. The value of Divergence analysis in fig.1b and its quantification fig.1c, fig.2c and its quantification Fig. 2d is not consistent. The range of the color scale may be wrong because the color bar is positive (like, 0~7), but the statistical values are negative (like, -0.08~0).

2. Extended fig.1d for border cell cross section, the image appears too blurred. It would be beneficial to obtain a higher resolution image to ensure clarity. Additionally, there seems to be an inconsistency in the MPAct/CaaX results compared to this representative image. For example, the MPAct-GFP signal in the center right is notably high, while the CaaX-Scarlet signal is quite low, yet the ratio is unexpectedly low in that region.

3. Fig.3a and the related movie (Supplementary Video 8) makes it challenging to compare the moving speed and persistence between the WT and mutant. To enhance clarity, it would be beneficial to include a smaller scale (fewer cells) or provide tracking markers to follow the movement more precisely. This adjustment would allow for a more accurate comparison of the dynamic behaviors between the two conditions.

4. The representative image of MoesinTD in Fig. 4f shows a rather large lamellipodium, which may not be typical. It would be more effective to replace it with a more representative image, as well as Supplementary Video 9.

5. Your discussion on the actin network's role to promote cell migration is thorough and insightful. To further enhance this section, you might consider referencing some recent work (Zhou et al., Two Rac1 pools integrate the direction and coordination of collective cell migration. *Nat Commun.* 2022; Georgantzoglou et al. A two-step search and run response to gradients shapes leukocyte navigation in vivo. *J Cell Biol.* 2022; Qian et al. Pulses of RhoA signaling stimulate actin polymerization and flow in protrusions to drive collective cell migration. *Curr Biol.* 2024), which explored how the actin flows regulating cell lamellar protrusion dynamics and migration behavior. This aligns closely with your observations and could provide additional context and support for your findings.

Version 1:

Reviewer comments:

Reviewer #1

(Remarks to the Author)

The authors have adequately addressed all my concerns.

Reviewer #2

(Remarks to the Author)

The authors have addressed my concerns and the paper is now ready for publication

Reviewer #3

(Remarks to the Author)

The study by Sánchez-Sánchez et al. beautifully demonstrated that *Drosophila* hemocytes utilize both lamellar and cortical actin networks during migration, with Moesin playing a crucial role in regulating their morphology and movement. This research revealed the coexistence and inter-regulation of these actin networks, highlighting Moesin's function in crosslinking actin filaments. These findings deepen our understanding of the complex mechanisms underlying cell migration. The revised manuscript fit the requirement for Nat Commun publication.

The revised version of Sánchez-Sánchez et al. has effectively addressed our concerns. However, we recommend that they should incorporate the data comparing Moesin and myosin mutant phenotypes (as presented in their response, Figures 4 and 5) into the supplementary figures of their manuscript and include these findings in the result section, and also discuss the aspect of myosin in the discussion section. Including all these would proactively address potential questions from readers regarding the role of myosin contractility, thereby reinforcing the conclusion that the observed effects on cell motility are mainly independent of myosin contractility.

Reviewer #1 (Remarks to the Author)

This manuscript by Sanchez-Sanchez et al., investigate the role of the actin cortex in the migration of *Drosophila* hemocytes in vivo. Using powerful microscopy experiments and analysis and different mutants of Moesin, sole member of the Ezrin-Radixin-Moesin family in flies, they show that the actin cortex is mainly restricted to the cell body and do not extend in the lamella, and that perturbing the cortex impairs migration. In addition, they demonstrate that Moesin is a key regulator of this process and that its function requires both binding to the actin cytoskeleton and to the plasma membrane via PIP2. This manuscript is a very comprehensive study that is well suited for publication in Nature Communications. It will be of interest to a broad audience. Indeed, even if its findings are not extremely surprising, the manuscript demonstrates that Moesin has an impact on the actin dynamics and their work suggests that cortical Moesin impacts on lamellae functions.

I have only minor concerns that are listed below:

1. Divergence in actin flow may not be clear for everyone (I needed a refresh). As such, the authors should explain the concept better. Also, they represent divergence on a scale of 0 to 7 or 0 to 12 in the figures which is different from their previous work (Yolland et al., 0 to -1). I suspect that the scale in the images should indeed be 0 to -1. If I am wrong and the scale is indeed correct, then the quantification provided (and the description in the text) makes little sense as it is in the negatives and does not compare to the respective images.

We thank the reviewer for this comment. We have now added some further information in the text (lines 56-60) and modified the figures (altering the divergence scale bar) and figures legend in Fig 1 and Fig 2 (lines 390-396, 416-421) to be consistent with our previous publication. The reviewer is correct that there was an error with the original scale and it should indeed be negative values as there is little positive divergence observed in the actin flow field.

2. In Fig.2, the authors found that lamellar size is unaffected, but the morphology of the lamella seems different in MoeMut (they seem shorter and thicker). It might be that this particular image is not representative of the phenotype, but, also when considering the results in Fig.7 (see below), a better characterisation of lamellae morphology to ensure that it is normal would reinforce the conclusions of the authors. This is particularly important as the authors assume for the rest of the paper that only cortical actin at the cell body is affected (again, see comments for Fig.7). The authors could for example determine the thickness, the length and the dynamics of the lamella in control and MoeMut.

The thickness of the lamella in a 3D reconstruction is very variable as the lamella is undulating as it shapes to the contour of the surface that it is in contact with. The apparently thicker view in this image seems to not be representative of the phenotype or even of other cross-sectional views of the same cell. To confirm that the lamella does not have an obvious morphological phenotype we have now measured lamellar volume and area and neither measurement suggest an alteration in the absence of Moesin

(Figure 1, response to reviewers). However, the leading edge lamellar dynamics are indeed affected (Figure 7, manuscript).

Figure 1. Measurement of lamellar and cell body volume and area in macrophages suggests that only the gross morphology of the cell body is affected in Moesin^{Mut}.

3. How does the authors explain that the MoeTAKN mutant expression lead to a phenotype as it is not associated with membrane nor actin ? Also that MoeTAKN seems cortical in Fig.5 ? This seems contradictory with their finding that both actin and membrane association are required for Moe accumulation at the cell body.

The image of the Moe^{TAKN} cell was not representative of the phenotype and this has now been changed (Figure 5A, manuscript). There does not appear to be a cortical enrichment of this mutant construct (Figure 5B, manuscript). However, we do observe a potentially subtle migration phenotype (Figure 4C, manuscript). There are several potential explanations for a subtle phenotype with this mutant. It is possible that there is still some residual actin or membrane binding in this point mutant, which could cause some dominant negative effect. It is difficult to rule out that these mutations completely prevent binding. Opening up of auto-inhibited Moesin is complex, and there is even a second hypothesized phosphorylation site that has been shown to play a role (10.1016/j.bpj.2017.10.041), which is unaffected in our mutant construct. Additionally, Moesin is thought to dimerize and it is possible that the point mutant is having a subtle effect by interacting with the endogenous wild-type protein.

4. Fig.6e the embryos are blurry (seems like a motion blur).

This image has now been replaced (Figure 6E, manuscript).

5. l.189-190: I don't think that the Morphotrap external can recruit a protein that is intracellular.

In the original morphotrap paper they show that the morphotrap external can indeed relocalize several membrane-associated proteins, presumably to the external face of the

lipid bilayer (<https://doi.org/10.7554/eLife.22549>). In contrast, the morphotrap internal localizes the protein to the internal face. We used the morphotrap external simply as a control UAS construct in this case as it should not be capable of correctly localizing the Moesin^{TDKN} to the internal face of the membrane to regulate the cortex. While it is possible that the morphotrap external does not work for Moesin as it did for the membrane proteins tested in the morphotrap paper, we believe that it is the best possible control for this experiment.

6. l.195: the conclusion that advection requires membrane binding could be reinforced by investigating the dynamics of the TDKN form with the Morphotrap internal.

This is a great suggestion, and we have discussed how to perform this experiment. However, there is not a free fluorescent channel to label the actin network. The morphotrap line is labelled with mCherry while the Moesin^{TDKN} is tagged with GFP, which is essential for the morphotrap to relocalize the mutant.

7. Fig.7 show that lamella is indeed affected when perturbing Moesin's activity. The authors conclude that the phenotype is basically due to the change at the cortex of the cell body. I am not sure about their argument there. Yes, most of Moe's role seems to be on the cell body, but there might be small amount of Moe in the lamella that might be sufficient to induce a phenotype. I am not sure how this could be tested (an optogenetic recruitment could work, but this seems to be excessively demanding at this stage), so if the authors cannot provide further data demonstrating that only cortical actin is perturbed, I would suggest that the author tone down their conclusion.

We completely agree with this point and have softened the interpretation of our results (Lines 202, 213-215, 262-272).

8. There is no "Acknowledgments" section.

We have now added an acknowledgement section (Lines 274-281).

Reviewer #2 (Remarks to the Author)

This beautiful study by Sanchez-Sanchez et al takes full advantage of the fly embryo and uses state of the art microscopy techniques to investigate the role of moesin in regulating the complex actin networks that drive macrophage migration in vivo. The imaging is exceptional and the authors use the powerful live imaging that the model system offers to characterise in unprecedented detail the organisation and dynamics of the actin network within fly macrophages (hemocytes) as they migrate within a developing fly. The authors show convincingly that these cells possess both a lamellar actin network at their front and a cortical actin network at the rear. Whilst the lamellar network has been previously studied the cortical network has not been looked at and consequently this study provides its first in depth characterisation which is an important piece of work. The fact that these cells present a hybrid actin organisation between a typically 'amoeboid' migratory cell and a 'mesenchymal' migratory cell is also fascinating. The authors show that the actin cortex regulator moesin plays an important role in regulating actin dynamics within the macrophages and intriguingly they show that both cortical and lamellipodial actin networks are inter-regulated. This is another fascinating observation and further work is required to work out the mechanism by which these two networks are coordinating their actions to drive migration.

Major comments:

I think to dissect out this mechanism would be beyond the scope of this work, however, in my opinion, focusing exclusively on these cells whilst they carry out developmental migration throughout the embryo misses an opportunity to expand the work to macrophage migration in a setting where they are carrying out a directed chemotactic migration to a focused source (for example a site of tissue injury or infection). Do the moesin mutants have a defect in this type of migration? In these settings how does the sensing of an external cue plug into the actin regulation that the authors observe in these cells to drive directed migration? In the majority of movies in the paper the cells are migrating randomly, how is moesin enabling the cell to polarise in response to an external signal (damage, infection, another cell to drive contact inhibition). If the authors can add data addressing this question to their study then I think it would have much wider appeal to the macrophage/immune cell migration community and that Nature Communication would be a perfect home for the paper.

These are interesting questions and we would indeed like to understand precisely how Moesin is enabling the cell to polarize in response to various cues. While we do observe a contact inhibition defect in the Moesin mutants (Figure 3D, manuscript) it is a little tricky to precisely dissect exactly what Moesin is needed for. The loss of Moesin activity in these cells leads to a pleiotropic effect that causes defects in cell polarity, cell speed, cell persistence, cell shape, actin dynamics, leading edge dynamics.... And there are several ways Moesin could be involved in the contact inhibition process as well as other external migratory cues.

We have now performed some analysis of macrophage responses to wounds and while the migration speed of Moesin RNAi expressing cells is reduced as expected, they do seem to still be able to respond. However, we do not yet know if their behaviors is affected at the wound site (e.g., phagocytosis defects) (Figure 2, response to reviewers).

Figure 2. (A) Quantification of macrophage recruitment to wounds. Macrophage numbers at the wound site were normalised to wound size to control for any variability in initial wound diameter. Over 30 minutes of imaging we did not observe a statistical difference in macrophage recruitment. (B) Quantification of macrophage migration speed during their wound recruitment reveals a statistically significant reduction in cells expression a moesin RNAi.

In addition, the following more minor points should be addressed:

1. Why is lamellar size unaffected in moesin mutants (Fig 2a,b) if moesin is required for the correct organisation of the cortical actin network and this network in turn regulates lamellipodial activity, would you not expect a difference in size as well as actin flow?

The size of a lamella will be governed by the rate of new addition of actin filaments at the leading edge minus the rate of retrograde flow. These two parameters need to be in balance if the lamella is to stay a constant size. What our data suggests, which is in line with recent work from others, is that there is feedback between contractile processes at the rear of the cell (or within the lamella) and activities at the cell front. As the lamella of Moesin mutants are of similar area to control cells there must be feedback involved to modulate the polymerization machinery at the leading edge of these cells. Changes in membrane tension are a recently hypothesized signal to coordinate such activities across the cell (10.1016/j.cell.2023.05.014).

2. The phenotypes with moesin mutants, moeRNAi, pten and skt1 mutants are all stronger than that observed with slik mutants (figure 3g) Can the authors show further evidence for a role of slik in regulating moesin in the macrophages? Do they see genetic interaction between slik and moesin for example? If slik is not regulating moesin phosphorylation than are there other candidates?

One possibility for the reduced phenotype in the slik mutant is that there are maternal levels of the protein still present at the time of embryogenesis in which these experiments were performed. We have now tested a slik RNAi line expressed in

macrophages and this leads to a phenotype with similar severity to the moesin mutant (Lines 106-107) (Figure 3, response to reviewers, and Figure 3G in the manuscript).

Figure 3. Quantification of macrophage developmental dispersal defects (also Figure 3G in the manuscript). Slik RNAi expression in macrophages leads to a similar phenotype to Moesin mutant or RNAi expressing cells.

3. Can the developmental migration defects be explained by a requirement for the actin cortex (and moesin) to allow the cell to withstand external forces coming from the spatial constraints experienced by macrophages as they carry out their migrations in the embryo?

Yes, this is a good point which we have added to the discussion (Lines 220-224).

4. This images of embryos in Figure 6e are out of focus and not up to the standard of the imaging throughout the rest of the paper. These need to be replaced.

These images have now been replaced (Figure 6E, manuscript).

5. The experiments using morphotrap internal and external are very elegant but beg the question - What is moesin binding to in the membrane? (i.e – what is the ligand depicted in Figure 4a within the migrating macrophages?). This may be beyond the scope of the current study but can the authors speculate?

This is an interesting point. However, the figure that we originally included may be misleading in that it is possible that no ligand is needed at all with regards to actin cortex regulation in these cells. Moesin is thought to bind the membrane directly through its PIP2 interaction and binding to actin filaments, which may allow it to directly regulate the cortex. There are examples where Moesin is indeed activated by a receptor mediated event and in this case a specific ligand is involved. We do not know, and do not currently hypothesize that there is temporal activation of Moesin through such a signaling event, although it is possible that a Vegf-like cue, PVF, could be involved as it is thought to play a role in macrophage dispersal. We have removed the ligand from the figure to remove any confusion (see change in Figure 4A, manuscript).

Reviewer #3 (Remarks to the Author):

Sánchez-Sánchez et al. reported an interesting and important finding about how lamellar and cortical F-actin networks link and crosstalk with each other in *Drosophila* macrophage migration. In this process, active moesin binding to plasma membrane can be advected to the rear by lamellar actin flow to control cortical F-actin network. Then, cortical actin network can facilitate actin flow and leading protrusion dynamic growth. Finally, authors showed that this moesin-mediated crosstalk between lamellar and cortical F-actin networks are critical for macrophage migration. This study provided novel aspects of the relationship between two different F-actin networks and highlights the importance of active moesin in this process and cell migration. Overall, this manuscript is well introduced, well-organized and main findings are well supported by their experimental results. A couple of important points needs to be addressed before the publication of this story in *Nat Commun*: for example, whether and how advected active moesin controls myosin assembly in cortical F-actin network; whether cell migration is mainly dependent on leading protrusion dynamic growth, or cortical actomyosin contraction, or both. Answering these important questions will definitely improve the quality of this manuscript.

Major comments:

- 1.** As previous study from this lab showed that myosin is also tightly related to actin flow in *Drosophila* macrophage (NCB 2019), it is unclear whether and how this advected/active moesin can affect myosin assembly at the cortical F-actin network. Authors should check myosin signals in the cortical region under some different genetic backgrounds.
- 2.** If myosin assembly is really affected by advected/active moesin, authors might need to check whether myosin contraction might be able to influence macrophage migration. Regarding that myosin contraction might affect several different aspects, experimental answering of myosin function on cell migration might be difficult. Authors can discuss the potential effects of actomyosin contraction in cortical F-actin networks, on macrophage migration.

Here we are responding points 1 and 2.

We have now examined the localization of Myosin by expressing a fluorescently tagged transgene of the Myosin-II heavy chain (*zip* in flies). We do not see a difference in the amount of Myosin surrounding the cell body in the absence of Moesin (Figure 4 A,B, response to reviewers). We have also co-localized the active form of Moesin (Moesin^{TD}) and Myosin (mCherry tagged myosin light chain; *sqh* in flies) and while there is some degree of overlap in regions of the cortex, it is not complete co-localization (Figure 4C, response to reviewers). Movies reveal that Myosin is extremely dynamic around the macrophage cortex while activated Moesin is always present (data not shown). Additionally, we do observe some differences in myosin and Moesin mutant phenotypes. As we previously revealed, myosin mutant macrophages have a highly unorganized actin flow field in their lamella as highlighted by streamline analysis (Figure 5, response to reviewers; 10.1038/s41556-019-0411-5). However, we do not observe a gross change in the organization of the flow field in the absence of Moesin (Figure 5, response to

reviewers). These data suggest that Moesin is likely not directly regulating myosin activity in these cells.

Figure 4. (A) Control and Moesin^{Mut} macrophages expressing Myosin-II showing enrichment at the cell cortex (arrows). (B) Quantification of Myosin-II enrichment at the cell cortex showing no statistical difference. (C) Example of cells co-expressing a single copy of Moesin^{TD} and Myosin-II at the cortex (arrows). Note that Moesin^{TD} has a more widespread localization surrounding the cortex compared to Myosin. Scale bars 10 μ m.

Figure 5. Actin distribution, PIV, and streamline analyses of actin flow in Myosin^{Mut} and Moesin^{Mut} macrophages. Actin flow organisation is perturbed in Myosin^{Mut} cells, however, Moesin Mutant shows a relatively coherent flow field, suggesting that these two proteins are playing relatively distinct roles in the lamella. Scale bar 10 μm .

3. It is unclear how advected/active moesin controls macrophage migration, although authors stated that moesin can affect retrograde actin flow and leading protrusion dynamic growth. Authors can discuss these several possibilities to explain how cell migration can be controlled by moesin.

Moesin likely has pleiotropic effects in the cell in regulating the actin cortex and potentially the actin network within the lamella. This regulation likely involves both direct interactions and indirect interactions as there is known feedback between these two actin networks. We have now expanded on this in the discussion (Lines 218-227).

4. Moesin^{TD} exhibits the most severe migration defects, and the authors analyzed the volume and also sphericity of cell body, and lamella, but did not show other phenotypes of others. Could authors analyze the same parameters of different mutants shown in Fig.4C?

We have analyzed morphological changes in all of the mutant lines in 2D and only Moesin^{TD} showed a lamellar phenotype with regards to changes in area (Figure 6, response to reviewers). Additionally, only Moesin^{TD} showed enrichment to the cortex (Figure 5B,C, manuscript) and a reduction in actin retrograde flow (Figure 7J, manuscript). We therefore focused on Moesin^{TD} for a more thorough morphological analysis in 3D, as the rest of the data suggested that if there were alterations in morphology in the other

genotypes they would be minor at best. The 3D analysis is time consuming and resource heavy and we therefore prioritized experiments to focus on Moesin^{TD}, which showed a highly penetrant genotype.

Figure 6. Measurement of lamellar and cell body area in macrophages suggests that only the gross morphology of the lamella is affected in Moesin^{TD} mutants.

Besides, authors showed a lot of quantification results of other two mutants, MoesinTA and MoesinTAKN, but authors did not mention the phenotypes/results of these two mutants (in figure 5, 6, 7) in the result texts.

We have altered the text to highlight that the phenotypes characterized in Figures 5-7 were specific to Moesin^{TD} compared to other transgenes (Lines 158-161, 167-170, 211-213).

5. Authors used 3D cell volume for quantification of cortical body. Is it possible that 2D area can also be used to present cell changes in cortical body? Is there some positive correlation between 3D cell volume and 2D cell area in cortical body? Authors should clarify this point.

The reviewer is correct in highlighting that 2D area can also be used to highlight differences in cortex morphology in Moesin Mutants (Figure 1, response to reviewers). However, for Moesin^{TD} we observed a statistically significant difference in 3D volume quantification (Figure 4E, manuscript). Yet, we do not observe a difference in the 2D projected area (Figure 6, response to reviewers). Therefore, the area and volume measurements may not always be correlated.

Minor comments:

1. The value of Divergence analysis in fig.1b and its quantification fig.1c, fig.2c and its quantification Fig. 2d is not consistent. The range of the color scale may be wrong because the color bar is positive (like, 0~7), but the statistical values are negative (like, -0.08~0).

Thank you for highlighting this mistake, which has now been corrected. See also response to Reviewer 1, point 1.

2. Extended fig.1d for border cell cross section, the image appears too blurred. It would be beneficial to obtain a higher resolution image to ensure clarity. Additionally, there seems to be an inconsistency in the MPAct/CaaX results compared to this representative image. For example, the MPAct-GFP signal in the center right is notably high, while the Caax-Scarlet signal is quite low, yet the ratio is unexpectedly low in that region.

Thank you very much for spotting this error. The original MPAct and plasma membrane images of the border cells were a maximum projection of the entire cluster, while the ratio was only performed on a slice, which was why the images were inconsistent. We have now changed the images (Figure S1D, manuscript) showing only the slice analyzed ratiometrically. The MPAct ratiometric analysis in border cells suggests that the cortical enrichment around the cluster is not homogenous and varies across a 3D stack.

3. Fig.3a and the related movie (Supplementary Video 8) makes it challenging to compare the moving speed and persistence between the WT and mutant. To enhance clarity, it would be beneficial to include a smaller scale (fewer cells) or provide tracking markers to follow the movement more precisely. This adjustment would allow for a more accurate comparison of the dynamic behaviors between the two conditions.

Sorry for any confusion. The analysis of persistence was not performed on the cells in Movie 8 as this analysis needs to be performed on cells imaged at a higher spatial and temporal resolution. Additionally, we need to only analyze cells that are freely moving and not undergoing contact inhibition as this would affect the persistence measurement. We have now included the tracks used to calculate persistence for an improved visualization (Figure 3C, manuscript).

4. The representative image of Moesin^{TD} in Fig. 4f shows a rather large lamellipodium, which may not be typical. It would be more effective to replace it with a more representative image, as well as Supplementary Video 9.

The reviewer is correct and this cell was not representative. We have changed the Moesin^{TD} example in Figure 4F in the manuscript and the associated movie (Supplementary video 9).

5. Your discussion on the actin network's role to promote cell migration is thorough and insightful. To further enhance this section, you might consider referencing some recent work (Zhou et al., Two Rac1 pools integrate the direction and coordination of collective cell migration. *Nat Commun.* 2022; Georgantzoglou et al. A two-step search and run response to gradients shapes leukocyte navigation in vivo. *J Cell Biol.* 2022; Qian et al. Pulses of RhoA signaling stimulate actin polymerization and flow in protrusions to drive collective cell migration. *Curr Biol.* 2024), which explored how the actin flows regulating cell lamellar protrusion dynamics and migration behavior. This aligns closely with your observations and could provide additional context and support for your findings.

Thank you for your suggestion. We have referenced the Georgantzoglou et al., paper which was indeed very relevant (Line 268).

REVIEWERS' COMMENTS

Reviewer #1 (Remarks to the Author):

The authors have adequately addressed all my concerns.

Reviewer #2 (Remarks to the Author):

The authors have addressed my concerns and the paper is now ready for publication

Reviewer #3 (Remarks to the Author):

The study by Sánchez-Sánchez et al. beautifully demonstrated that *Drosophila* hemocytes utilize both lamellar and cortical actin networks during migration, with Moesin playing a crucial role in regulating their morphology and movement. This research revealed the coexistence and inter-regulation of these actin networks, highlighting Moesin's function in crosslinking actin filaments. These findings deepen our understanding of the complex mechanisms underlying cell migration. The revised manuscript fits the requirement for *Nat Commun* publication.

The revised version of Sánchez-Sánchez et al. has effectively addressed our concerns. However, we recommend that they should incorporate the data comparing Moesin and myosin mutant phenotypes (as presented in their response, Figures 4 and 5) into the supplementary figures of their manuscript and include these findings in the result section, and also discuss the aspect of myosin in the discussion section. Including all these would proactively address potential questions from readers regarding the role of myosin contractility, thereby reinforcing the conclusion that the observed effects on cell motility are mainly independent of myosin contractility.

We thank the reviewer for this comment. We have combined Figures 4 and 5 as a new Supplementary Figure 3. We have added these findings in the result section (Lines 218-226) and the discussion (Lines 277-279).